

# Consistency assessment of rating curve data in various locations using Bidirectional Reach (BReach)

Katrien Van Eerdenbrugh[1], Stijn Van Hoey[2], Gemma Coxon[3], Jim Freer[3], and Niko E.C. Verhoest[1]

[1]Laboratory of Hydrology and Water Management, Ghent University, Coupure Links 653, B-9000 Ghent, Belgium
[2]Research Institute for Nature and Forest (INBO), Kliniekstraat 25, B-1071 Brussels, Belgium
[3]School of Geographical Sciences, University of Bristol, University Road, Bristol, United Kingdom

*Correspondence to:* Katrien Van Eerdenbrugh (katrien.vaneerdenbrugh@ugent.be)

**Abstract.** When estimating discharges through rating curves, temporal data consistency is a critical issue. In this research, consistency in stage-discharge data is investigated using a methodology called Bidirectional Reach (BReach), which departs from a (in operational hydrology) commonly used definition of consistency. A period is considered to be consistent if no consecutive and systematic deviations from a current situation occur that exceed observational uncertainty. Therefore, the
capability of a rating curve model to describe a subset of the (chronologically sorted) data is assessed in each observation by indicating the outermost data points for which the rating curve model behaves satisfactory. These points are called the maximum left or right reach, depending on the direction of the investigation. This temporal reach should not be confused with a spatial reach (indicating a part of a river). Changes in these reaches throughout the data series indicate possible changes in data consistency and if not resolved could introduce additional errors and biases. In this research, various measurement stations
in the UK, New Zealand and Belgium are selected based on their significant historical ratings information and their specific characteristics related to data consistency. For each country, regional information is maximally used to estimate observational uncertainty. Based on this uncertainty, a BReach analysis is performed and subsequently, results are validated against available knowledge about the history and behavior of the site. For all investigated cases, the methodology provides results that appear consistent with this knowledge of historical changes and facilitates thus a reliable assessment of (in)consistent periods in stage-
discharge measurements. This assessment is not only useful for the analysis and determination of discharge time series, but also to enhance applications based on these data (e.g., by informing hydrological and hydraulic model evaluation design about consistent time periods to analyze).

## 1  Introduction

For many applications in hydraulics, hydrology and water management, reliable river discharges are crucial. A commonly used
practice for the estimation of these discharges is the use of rating curves. Through the calibration of a relation between stage and discharge measurements (i.e. a rating curve), high-frequency stage measurements can be transformed into high-frequency discharge measurements. As this relation is based on only a limited number of simultaneous stage-discharge measurements, it is a relatively budget-friendly method for discharge assessment in rivers.



The use of rating curves requires attention for the consistency of the measured stage-discharge data set. Several causes (e.g., geometric changes of the river bed, infrastructure works, weed growth) can alter the hydraulic behavior of the river in the considered measurement location temporarily or permanently and thus limit the validity of a calibrated rating curve. Information about this temporal (in)consistency is hence critical to prevent additional errors and biases from occurring in the

determined river discharges. Moreover, a correct assessment of (in)consistent periods can enhance other applications based on the investigated data. For instance, errors in hydrological or hydraulic model results that are caused by changes in a river's situation can (if they lead to inconsistency of the rating curve data as well) be avoided by using known consistent stage-discharge time periods for model evaluation. Methods to detect and describe this temporal (in)consistency have been studied by several authors (see Van Eerdenbrugh et al. (2016) for a more extensive review). McMillan et al. (2010) assumed that

changes in rating curve behavior are mainly caused by floods and that periods in between are consistent. Other methods described the variation of rating curve parameters in time (Westerberg et al., 2011; Reitan and Petersen-Øverleir, 2011). Within predefined consistent periods, Jalbert et al. (2011) accounted for an aging error toward an initial rating curve, which expresses the increasing risk of a change of the river bed in time. Morlot et al. (2014) expanded this method with two preliminary steps. First, the stage-discharge data set is segmented into consistent periods and subsequently hydraulic analogues of each stage-

discharge measurement are selected within these periods. Although measurement uncertainties are considered in the latter step, the first and thus defining data segmentation does not account for them.

Van Eerdenbrugh et al. (2016) discuss that in the methods mentioned above, the assessment of temporary or permanent changes of the hydraulic regime requires assumptions or decisions that more or less influence eventual results. Moreover, most of the methods start from a definitive choice of a rating curve model and comprehend an assessment of its parameters

distribution. However, if data consistency is assessed prior to a definitive in-depth analysis (as in Morlot et al. (2014)), it can provide an increased understanding that contributes to the selection of an appropriate, more definitive rating curve model. An important criterion for this preliminary consistency analysis is that results minimally depend upon choices and decisions made by users.

Therefore, Van Eerdenbrugh et al. (2016) have developed a methodology to enable the detection of consistent periods in

stage-discharge data. It is called Bidirectional Reach (BReach) and considers a period to be consistent if no consecutive and systematic deviations from a current situation occur that exceed observational uncertainty. This definition of consistency is commonly used in operational hydrology (Reitan and Petersen-Øverleir, 2011). It requires the assessment of (1) observational uncertainty, (2) a current situation and (3) the consecutive and systematic character of nonacceptable deviations. Observational uncertainty is estimated for each country using regional information (Sect. 2.2.3 and 2.4). The assessment of a current situation

is done by evaluating the capability of a rating curve model to describe a subset of the data in each pair of stage-discharge measurements (Sect. 2.2.5). This capability is defined by a degree of tolerance, i.e. a definition of satisfactory behavior for the rating curve model in a series of gauging points (Sect. 2.2.4). By combining multiple degrees of tolerance, complementary information is provided that allows for the exclusion of causes for model failure other than data inconsistency. Hence, changes throughout time in the combined model performance indicate possible changes in data consistency. This information is used

for the last requirement in the definition of consistent periods, i.e. the assessment of the consecutive and systematic character of



nonacceptable deviations (Sect. 2.2.6). In Sect. 2, the different steps of the methodology are briefly explained and all necessary choices are discussed.

In Van Eerdenbrugh et al. (2016), one observed and several synthetic data sets are used to evaluate and test the robustness of the methodology. The method was shown to perform well with robust results despite decreased data availability, erroneous estimations of measurement uncertainty and even a partially deficient rating curve model. As all investigated data sets belong to the same geographical location, an additional analysis with more diverse measured data sets is important to further explore the methodology's applicability. In this paper, several gauging stations in the United Kingdom (UK), New Zealand and Belgium are selected for this purpose based on their well-documented history and their specific characteristics related to rating curve consistency. For each country, regional information is maximally used to estimate observational uncertainty. Based on this uncertainty, a BReach analysis is performed and subsequently, results are validated against available knowledge about the history and behavior of the site.

## 2    Methods

### 2.1    Study areas and data

The BReach methodology is applied to three stage-discharge data sets in the UK, two in New Zealand and five in Belgium. These stations are selected based on their particular properties with regard to data consistency. Their well-documented history enables a verification of the results of a BReach analysis. An overview of these stations and their main characteristics is given in Table 1. The UK data are provided with a quality indication and hence only stage-discharge measurements marked as 'good' are used in this research. The New Zealand data were preprocessed by the Horizons Regional Council and the Marlborough Regional Council and are assumed to have a sufficient quality level. For the Belgian stations, raw (unprocessed) gauging data are available. Therefore, stage-discharge measurements with recorded stages that deviate more than 5 cm from the nearest continuous value are treated as outliers and not used in the analysis. These continuous stage data have a temporal resolution of one hour (before 2003) and of 15 minutes (after 2003). Taking into account the estimated 95 % uncertainty boundaries of the (gauging) stage measurements ($\pm 2$ cm) and assuming a similar magnitude for those of the continuous measurements (Sect. 2.4.3 and 2.2.3), this difference of 5 cm guarantees that only measurements with large errors are excluded from the analysis.

### 2.2    BReach methodology: description and practical application

The aim of the BReach methodology is to identify consistency in rating curve data based on a quality analysis of model results. The methodology consists of several consecutive steps (Van Eerdenbrugh et al., 2016):

Step 1:  Selection of a model structure for the analysis;

Step 2:  Sampling of the parameter space;

Step 3:  Assessment of acceptable model results;





Step 4:  Assessment of different degrees of tolerance;

Step 5:  Assessment of the bidirectional reach for all degrees of tolerance;

Step 6:  Identification of consistent data periods.

In this section, all steps and their practical application in this paper are briefly described.

### 2.2.1  Step 1: Selection of a model structure for the analysis

A first step is the choice of a rating curve model that appropriately approximates the relation between discharge and stage for an important part of the measured range. In this paper, the chosen rating curve model depends on the characteristics of the measurement station.

For the station of Colsterworth (UK, Table 1), a flat V-weir controls the flow and thus a power law can be used to describe the stage-discharge relationship:

$$Q = c(h - h_0)^n \tag{1}$$

where $Q$ is the discharge [m$^3$ s$^{-1}$], $c$ is a scale coefficient [m$^{3/n}$ s$^{-1}$], $h$ is the stage [m], $h_0$ is a location parameter [m] that expresses the stage of zero flow and $n$ is an exponent [-] that is a function of the type and the shape of the considered cross section.

For all other analyzed stations (except from the station of Clog-y-Fran, UK), a segmented rating curve with two segments is used (e.g., Reitan and Petersen-Øverleir, 2008; Le Coz et al., 2014; McMillan and Westerberg, 2015):

$$Q = \begin{cases} c_0(h - h_0)^{n_0} & h < h_{br,1} \\ c_1(h - h_1)^{n_1} & h \geq h_{br,1} \end{cases} \tag{2}$$

where each segment describes a different flow situation and $h_{br,1}$ is the breaking point between two consecutive segments. In this breaking point, continuity between both segments must be provided. By using this rating curve model with a breaking point at low flow conditions, it is possible to account for two different situations. First, the model is able to describe a change in flow situation. In many cases, flow at low stages is locally controlled (e.g., by one or several riffles). At higher stages, the flow situation at these riffles becomes drowned and the flow is controlled by a longer river reach (e.g., Reitan and Petersen-Øverleir, 2008; Le Coz et al., 2014). Second, a two segmented rating curve allows to account for the effect of geomorphological changes throughout time. If the river bed deepens, the value of $h_0$ in eq. (1) is expected to decrease. It is thus possible that in certain periods of the measured stage-discharge data, the stage of zero flow is higher than the lowest measured stage within the complete period and hence the sampling range of $h_0$ (Sect. 2.2.2) is too narrow. For these periods, the use of a second segment can overcome this shortcoming and the role of the first segment will thus be small(er).

Although both model structures are simple, this approach is satisfactory for nearly all stations. By analyzing well-chosen subsets of the data (e.g., winter data if influence of weed growth can be expected as in the river Grote Nete at Hulshout,




Belgium) or by performing an analysis on the data after sorting them by stage instead of chronologically (e.g., to assess the influence of a downstream movable weir in the river Meuse at Maaseik, Belgium), the chosen rating curve models also satisfy for less straightforward flow situations (cfr. Sect. 2.3, 3.7 and 3.8).

The complex flow situation in the river Taf at Clog-y-Fran (UK) however requires a rating curve model with increased
complexity. In this station, the hydraulic behavior is influenced by the combination of weed growth affecting low flow behavior, a considerable overspill over the right bank at higher stages and an unstable bed control. For these reasons, a segmented rating curve with three segments is used. The second segment overcomes similar difficulties as described for the two-segmented rating curve. The third segment is representing the flow for stages higher than bank overspill.

### 2.2.2   Step 2: Sampling of the parameter space

The sampling of the power law parameters (Eq.1) is nearly similar to Van Eerdenbrugh et al. (2016) where sampling intervals are bounded to a physically realistic order of magnitude. $h_0$ is sampled from the interval [$h_{bed}$ - 40 cm, $h_{min,cont}$ - 2 cm], where $h_{bed}$ is the lowest bed level of the available reliable cross section measurements. If no cross section data are available, it is the local datum toward which the measured stages are expressed. The value of $h_{min,cont}$ is the lowest measured stage in the continuously measured data series. Samples of $n$ are taken from the interval [0.5, 3.5]. The outermost values obtained
when applying the power law function for all gauging points with the upper and lower limits for $h_0$ and $n$ are used to define the sampling interval for the coefficient $c$. The lower limit is obtained by halving the resulting lowest value for $c$ and for the upper limit, the highest value is doubled. Both parameters $h_0$ and $n$ are sampled from a uniform distribution. For parameter $c$, a more dense sampling is aimed at for smaller values. Hence, this parameter is sampled from a uniform distribution after log-transformation.

The two-segmented rating curve (Eq.2) has 7 parameters, of which $h_1$ is computed to obtain continuity between the two consecutive segments of the rating curve. Sampling of $h_0$, $n_0$, $n_1$, $c_0$ and $c_1$ is the same as for the single power law. The sampling interval for $h_{br,1}$ is [$h_{min,cont}$ - 2 cm, $h_{br,max}$]. For all stations, the height range in which the lowest flows occur is assessed visually from the stage-discharge plots. An upper limit of this height range is estimated and taken as a value for $h_{br,max}$ (Table 3).

The three-segmented rating curve used at Clog-y-Fran has 11 parameters, of which $h_0$, $n_0$, $n_1$, $n_2$, $c_0$, $c_1$ and $c_2$ are sampled similarly as for the two-segmented power law and $h_1$ and $h_2$ are again computed to obtain continuity between two consecutive segments. Based on the stage-discharge data, the sampling interval for $h_{br,1}$ is chosen [$h_{min,cont}$ - 2 cm, 1 m]. Based on the information about out-of-bank flow at higher stages, $h_{br,2}$ is sampled from the interval [2.9 m, 3.5 m]. Both parameters are sampled from a uniform distribution.

For all types of rating curves, the parameter space is sampled using a Latin Hypercube sampling. For the single power law, $1.3 \times 10^6$ samples are taken. For the two-segmented and the three-segmented power law, $6.5 \times 10^6$ respectively $1.3 \times 10^7$ samples are taken.





### 2.2.3 Step 3: Assessment of acceptable model results

Following Van Eerdenbrugh et al. (2016), a result of a rating curve model and a parameter set is classified as acceptable if it fits
in a rectangular acceptance zone that is enclosed by the 95 % uncertainty boundaries of the accompanying stage and discharge
measurement.

An estimation of these measurement uncertainties is made by many authors. A good literature overview with a summary
of the major findings is given in Pelletier (1988) and in McMillan et al. (2012) for both stage and discharge measurements.
In these studies, errors on stage measurements are generally indicated as relatively small. Most of the estimated 95 % un-
certainty boundaries lie within $\pm 10$ mm, although values up to $\pm 40$ mm are also mentioned for more uncertain locations.
Error distributions are mostly assumed to have a negligible bias and to be independent of the value of the measured variable
(homoscedastic).

Discharge measurements are more uncertain and their errors are subjected to heteroscedasticity, i.e. error distributions vary
with changing discharge values. Therefore, uncertainty on discharge measurements is typically expressed as a percentage of
the occurring discharge. In nearly all studies, it is assumed to have a negligible bias. Pelletier (1988) reports 95 % uncertainty
boundaries between $\pm 4$ % and $\pm 17$ % for 5-35 verticals with the velocity-area method. McMillan et al. (2012) report the same
order of magnitude for the velocity-area method with various techniques. Coxon et al. (2015) found that despite expressing
errors as a percentage of occurring discharge, the value of the scale parameter in the assumed error distribution depends on the
value of the normalized discharge (i.e. measured discharge divided by mean discharge) with 95 % boundaries up to $\pm 25$ % for
low normalized flows and $\pm 13$ % for the highest normalized flows.

Although most of these studies share some general considerations, eventual uncertainties on stage and discharge measure-
ments can depend upon location, flow conditions and measurement technique and hence estimated uncertainty boundaries are
subjected to a relatively large variation. Therefore, this paper maximally uses available local information for the estimation
of observational uncertainty boundaries. Nevertheless, the ranges provided in literature offer a valuable framework to validate
these local findings. For each country separately, an estimation of local uncertainty boundaries is described in Sect. 2.4.

Based on the estimated observational uncertainty, results of a model (i.e. a rating curve model with a sampled set of param-
eters) are categorized as acceptable or nonacceptable. The result of this step is a binary matrix with classification results for
each parameter set and each data point.

### 2.2.4 Step 4: Assessment of different degrees of tolerance

As mentioned in the introduction, the BReach methodology evaluates the capacity of a rating curve model to describe a subset
of the data in each observation. For this evaluation, a definition of satisfactory behavior of a rating curve model is necessary.
In this paper, this definition is called the degree of tolerance and expresses the percentage of model results that are allowed to
be nonacceptable in a sequence of data points.

Van Eerdenbrugh et al. (2016) discuss that possible causes for a rating curve model to be nonacceptable in a data point are
(1) the occurrence of a higher observational error than estimated for the definition of acceptable results, (2) model deficiency





in certain ranges of the investigated variables and (3) data inconsistency. Due to the random occurrence in time of causes (1) and (2), corresponding nonacceptable results tend to be singularities in a chronologically sorted series of stage-discharge measurements. If on the contrary a nonacceptable model result is caused by the occurrence of data inconsistency, it can be expected that for the same model, nonacceptable results will also occur in other, neighboring (in time) data points. Hence,

these causes of failure will be highlighted when using higher degrees of tolerance (i.e. relaxation of the amount of points that can be nonacceptable in a sequence of data). As different degrees of tolerance provide complementary information, degrees of 0 %, 5 %, 10 %, 20 % and 40 % are used for this research.

### 2.2.5 Step 5: Assessment of the bidirectional reach for all degrees of tolerance

Before assessing the bidirectional reach, all stage-discharge measurements are sorted chronologically and their index within this

sorted data series is used to refer to them. Subsequently, a degree of tolerance is selected and the binary classification matrix is used to evaluate a model and its results from the perspective of one data point. The temporal span for which this model behaves satisfactory is assessed both in the direction of the previous and the following data points. Within these spans, the index of the outermost observation with an acceptable result is referred to as the left (previous points) or right (following points) reach. This information is aggregated for all parameter sets by taking the outermost left and right reaches. They are called the maximum

left and right reach and represent the indices beyond which none of the sampled parameter sets is acceptable within a data series with satisfactory behavior. Assessment of the maximum left and right reach is repeated for all data points and for all degrees of tolerance and results are summarized in a combined BReach plot (e.g., Fig. 3a). In this plot, each gray tint represents results for a specific degree of tolerance. For each data point in the x axis, the gray zone represents the span between the index of the maximum left reach (under the bisector) and the maximum right reach (above the bisector). The vertical distance between

the bisector and the index of the maximum left reach represents the maximum amount of data points before the investigated data point that can be described with at least one set of parameters under the chosen degree of tolerance. Similarly, the vertical distance between the index of the maximum right reach and the bisector represents this maximum amount for the data points after the investigated data point.

### 2.2.6 Step 6: Identification of consistent data periods

Combined BReach plots (e.g., Fig. 3a) provide a visual means to evaluate the capability of the rating curve models to describe a subset of the data in each point. Changes of this capability throughout time result in discontinuities of a BReach plot and each degree of tolerance provides complementary information. In accordance with the discussion in Sect. 2.2.4, discontinuities in the maximum reaches for stringent degrees of tolerance provide information about the diversity of measurements caused by (1) the occurrence of a higher observational error than estimated for the definition of acceptable results, (2) model deficiency in certain

ranges of the investigated variables, or (3) data inconsistency. The resulting BReach plots show changes in model performance precisely, but include too wide a variety of possible causes to detect data inconsistency. For a higher degree of tolerance, a model is allowed to generate nonacceptable results in a larger percentage of the data points. Therefore, discontinuities caused by (1) and (2) will disappear from the plots due to their random character. As a result, changes in consistency will be emphasized




in the plot but the larger tolerance does not facilitate a precise location of these changes. If plots that combine all degrees of tolerance indicate consistent data periods (i.e. periods without important discontinuities), plots with higher degrees of tolerance are used to assess the amount and indicative locations of consistency changes and based on this information, plots with stringent degrees of tolerance are used to locate these possible consistency changes more precisely (Van Eerdenbrugh et al., 2016).

## 2.3 Alternative analyses

In this paper, a BReach analysis is performed for all stations. If a seasonal variation in the rating curve behavior (due to weed growth) is presumed, a second analysis is performed on a subset of data measured during winter months (between December and March). In the UK and in Belgium, such a set of winter data is not expected to be influenced by weed growth. The combination of a BReach analysis on all data that shows no consistency and an analysis on only winter data that indicates consistent periods can confirm the influence of weed growth.

If it can be assumed that the behavior of the rating curve changes with changing stages, an additional BReach analysis is performed. For the latter, the data are sorted by stage instead of chronologically. Results of such an analysis can reveal in which height ranges the rating curve behavior alters. As multi-segmented rating curves aim to overcome these alterations, it is not interesting to use them in this context. Therefore, a single power law is used for all BReach analyses on data sorted by stage.

To avoid confusion between both a temporal BReach analysis and an analysis on data sorted by stage and between several types of rating curve models, results of the analyses will be referred to as $BReach_{x\_ys}$. In this formulation, $x$ is the type of analysis ($t$ (on chronologically sorted data) or $h$ (on data sorted by stage)) and $y$ is the amount of segments in the chosen rating curve model.

## 2.4 Assessment of uncertainties on stage and discharge measurements

The assessment of 95 % uncertainty boundaries of the stage and discharge data is based on available local information. This information availability differs for each country and hence in this section, the followed approach is described per country.

### 2.4.1 UK measurement stations

For the UK stations, Coxon et al. (2015) have analyzed the relative rating curve residuals from 26 measurement stations with very stable rating curves. A relative residual is defined as the ratio of the deviation (between discharge measurement and derived rating curve) and the measured discharge. The distribution of these residuals is investigated for different bins of normalized flow $Q_n$ (i.e. measured flow divided by mean flow). Results of this investigation show that logistic distributions with a zero location parameter (i.e. $\mu = 0$) and a scale parameter ($\sigma$) that varies exponentially with normalized discharge (Eq.(3)) fit the residuals well for all bins.

$$\sigma = 4.18e^{(-3.051Q_n)} + 3.531 \tag{3}$$

The 95 % uncertainty boundaries of discharge measurements for the UK data used in this paper are derived from these distributions and vary between ±28 % for the lowest normalized flows and ±13 % for the highest normalized flows. For stage



measurements, that typically have smaller measurement errors than discharges, a uniform error of ±5 mm was assumed by Coxon et al. (2015). Again, 95 % boundaries of this error (±4.875 mm) are used for the definition of the acceptance zone in the BReach methodology.

### 2.4.2 New Zealand measurement stations

In McMillan et al. (2010), the uncertainty on measured discharges in the measurement station of Barnett's Bank is assumed to follow a Gaussian distribution with zero mean ($\mu$) and a standard deviation ($\sigma$) of 4 %. Errors on stage measurement are considered Gaussian with zero mean and a standard deviation of 2 cm. 95 % uncertainty boundaries are thus ±8 % for discharges and ±4 cm for stages. These estimations are based on literature data and local expertise.

However, McMillan and Westerberg (2015) assume error distributions for Barnett's Bank similarly as described in Sect.
2.4.1 (Coxon et al., 2015). In this case, 95 % uncertainty boundaries for discharge measurements vary again between ±28 % for the lowest normalized flows and ±13 % for high normalized flows and are thus substantially higher than in the above mentioned approximation with a normal distribution. Stage uncertainty boundaries on the contrary are estimated smaller by Coxon et al. (2015) (±4.875 mm versus ±4 cm). Therefore, two different BReach analyses are performed for all New Zealand data, each based on one of these uncertainty estimations and results are compared.

### 15 2.4.3 Belgian measurement stations

For the Belgian measurement stations, no prior information concerning measurement uncertainties was available. Nevertheless, it is possible to gain insight in plausible characteristics of measurement errors by analyzing simultaneous measurements. Although in this paper, a BReach analysis is performed on only five Belgian measurement stations, simultaneous measurements of nine different stations are used for a preliminary uncertainty assessment of discharge measurements in order to maximize
the amount of (scarce) data.

A pair of simultaneously measured discharges consists of two discharge measurements that are measured with the same type of device within a time span of two hours and for which the corresponding measured stages are identical. Combining this information for 9 Belgian stations results in a set of 42 simultaneous pairs that are all measured with an OTT QLiner. The restriction to only one type of measurement device prevents a mixture of possibly different error distributions, each
corresponding with a different measurement technique. The errors of two simultaneous measurements are assumed to be independent.

To overcome the heteroscedastic character of discharge measurement errors, they are expressed as a percentage of the real discharge. Nevertheless, different authors find that parameters of errors distributions change with changing discharges (e.g., McMillan et al., 2012; Coxon et al., 2015). To investigate this, the simultaneous discharge measurements are sorted according
to their normalized discharge (cfr. Sect. 2.4.1). Subsequently, two subsets of this dataset are created, containing the 21 lowest respectively highest pairs of measurement. They are referred to as low flow and high flow data. These subsets are assumed to be unbiased (error distribution with zero mean, cfr. Sect. 2.2.3).





Both data sets do not allow for a direct assessment of measurement errors. However, if an error distribution is assumed, it is possible to test equality between the distributions of both the relative differences of the simultaneously measured discharge pairs and a created set of relative differences based on two equally sized samples of measurement errors from the assumed distribution. For instance, a Gaussian measurement error with zero mean and a standard deviation of 4 % is assumed for the low flow data set (cfr. Sect. 2.4.2 and McMillan et al. (2010)). From this distribution, two samples $\epsilon_1$ and $\epsilon_2$ are taken, each with size $m$ (in this paper $m = 10^6$) and they pairwise represent the assumed errors of two simultaneous flow measurements. As these errors are expressed as a percentage of the real discharge, a measurement (for both $j = 1$ and $j = 2$) can be written as:

$$Q_{meas,j,i} = (1 + \epsilon_{j,i})Q_{true,i} \tag{4}$$

with $i \in [1,m]$, $Q_{meas,j,i}$ one of both measured discharges in measurement pair $i$ and $Q_{true,i}$ the real discharge that occurred during the measurements. Combining Eq. (4) for both measurements in a pair leads to:

$$\frac{Q_{meas,1,i} - Q_{meas,2,i}}{Q_{meas,1,i}} = \frac{\epsilon_{1,i} - \epsilon_{2,i}}{1 + \epsilon_{1,i}} \tag{5}$$

Independently of the real discharge, this relative difference of two simultaneous measurements can thus be expressed by their measurement errors. If the assumed error distribution (Gaussian, $\mu = 0$ % and $\sigma = 4$ %) is correct, a data set calculated from the measurements pairs using the left-hand side of Eq. (5) (further called $\Delta Q_{m,\%}$) will have the same distribution as a data set calculated from the two sets of sampled errors using the right-hand side of Eq. (5) (further called $\Delta Q_{\epsilon,\%}$). When applying a two-sample nonparametric Kolmogorov-Smirnov test (KS test) on these data sets, the resulting p-value is 0.62, which is much higher than the commonly used 5 % level for rejection of the hypothesis that both data sets are equally distributed. The corresponding value of the Kolmogorov-Smirnov statistic (KS statistic) is 0.16. This is the maximum vertical distance between the empirical cumulative distribution functions (ECDF) of both tested data sets (Fig. 1a).

The same analysis is repeated for both low and high flow data and for Gaussian and logistic error distributions with different values of the scale parameters, equidistantly taken from the interval [1 %, 6 %] and [0.35 %, 4.4 %], respectively. As an example, Fig. 1b shows the resulting values of the KS statistic against the corresponding value of the scale parameter for the low flow data set using a Gaussian distribution. A p-value resulting from a KS test depends both on the value of the KS statistic and on the number of points in the investigated data sets. As the latter remains constant for all tests, p-values and values of the KS statistic will show a similar (although inverse) pattern. Fig. 1b clearly shows the occurrence of the lowest value of the KS statistic (and corresponding highest p-value) for a standard deviation of 3.12 %. In Fig. 1c, the ECDF of $\Delta Q_{m,\%}$ corresponds well with the ECDF of $\Delta Q_{\epsilon,\%}$ for this latter distribution. However, the occurrence of a high p-value (and corresponding small value of the KS statistic) provides no confirmation of the null hypothesis and it is possible that many other hypotheses lead to similar p-values. Nevertheless, the value of the KS statistic provides information not only about differences in central tendency but about any difference in the ECDFs. From this perspective, Spear and Hornberger (1980), Hornberger and Spear (1981) and Hornberger et al. (1985) compared the ECDFs of both behavioral and nonbehavioral parameter values and used the KS statistic as a measure for the sensitivity of a parameter. In this research, it is used to evaluate the behavior of error distributions



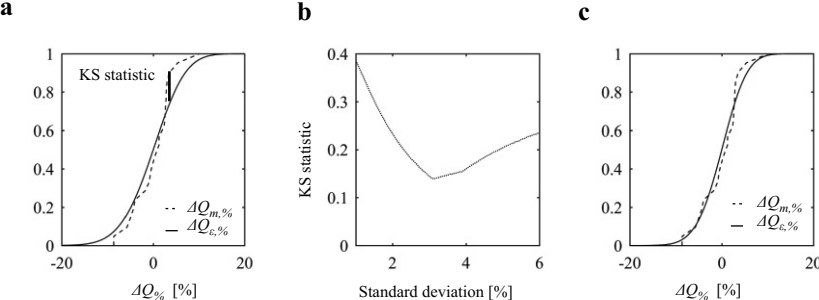

**Figure 1.** (a) ECDF and corresponding KS statistic of both $\Delta Q_{m,\%}$ and $\Delta Q_{\epsilon,\%}$ with $\mu = 0$ % and $\sigma = 4$ %. (b) KS statistics with $\mu = 0$ % and different values for $\sigma$. (c) ECDF of both $\Delta Q_{m,\%}$ and $\Delta Q_{\epsilon,\%}$ with $\mu = 0$ % and $\sigma = 3.12$ %. All plots are made for low flow data and a Gaussian error distribution.

that are a priori chosen based on currently available knowledge, without excluding the plausibility of other, unexplored error distributions.

The pattern of the other results (Gaussian distribution with high flow data, logistic distribution with both low and high flow data) is very similar. Table 2 shows the characteristics of both flow classes and both distribution types that correspond with a minimum KS statistic. There is a good correspondence between the ECDFs of $\Delta Q_{m,\%}$ and $\Delta Q_{\epsilon,\%}$ for both distribution types. Adjusted p-values (i.e. p-values after Benjamini-Hochberg correction, that accounts for a false discovery rate (Benjamini and Hochberg, 1995)) and KS statistics are also very similar and prohibit hence to make a distinction in favor of one single distribution type. Likewise in other studies (Sect. 2.2.3), this table clearly indicates that high flow data correspond with lower values of the scale parameters (and thus smaller uncertainty boundaries) than low flow data. For each flow class, the 95 % uncertainty boundaries of the two distributions do not strongly differ, but they are relatively small compared with the uncertainty boundaries applied in Sect. 2.4.1 and 2.4.2 and with literature data (e.g., Pelletier, 1988; McMillan et al., 2012). A tentative explanation for these low uncertainty values could be the relatively tranquil flow situations in the investigated measurement stations due to low slopes. Moreover, most of the investigated locations are situated at a bridge, facilitating discharge measurements in controlled conditions.

The limited amount of data prohibits a more precise description of this tendency toward lower uncertainties for higher normalized discharges. Moreover, the lowest normalized flow in the set of simultaneous discharge measurements is 0.72. Results of Coxon et al. (2015) show that an increase of measurement uncertainties can be expected for lower normalized flows. As more than 80 % of all investigated Belgian stage-discharge data have normalized discharges within the range of the low flow data subset or lower, it was decided to assume 95 % uncertainty boundaries to be $\pm 6.4$ % for all investigated Belgian discharge measurements. Although these values originate from the investigated low flow data measured with QLiners, they are applied for all discharge measurements, independent of their measurement technique. It can be expected that discharge uncertainties will differ for different techniques (e.g., McMillan et al., 2012; Song et al., 2012; Le Coz et al., 2014), but a





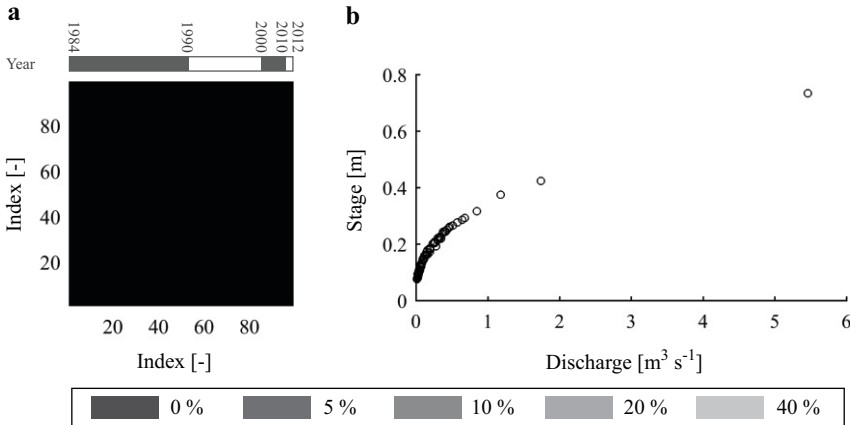

**Figure 2.** (a) Combined BReach$_{t\_1s}$ plot (all data) and (b) stage-discharge data for Colsterworth.

lack of simultaneous discharge measurements prohibits to perform a similar analysis for other measurement devices. Extra measurement campaigns might augment insight for these devices.

For the assessment of uncertainties on stage measurements, the data availability is different. Two simultaneous stage measurements are provided for each stage-discharge measurement. However, the first type of measurement is recorded from a staff gauge during a discharge measurement and the second type is registered by a continuous measurement device. Hence, it can be expected that error distributions of both data types differ and a similar approach as for discharge measurements is not justified. Therefore, 95 % uncertainty boundaries are estimated to be ±2 cm. This value is based on literature data and on local expertise.

## 3 Results and discussion

For each measurement station, results of the BReach analyses are validated using the available local information.

### 3.1 River Witham at Colsterworth, UK

In Fig. 2a, a combined BReach$_{t\_1s}$ plot is shown for Colsterworth. The BReach plot shows data consistency during the entire measured period. This corresponds with the nature of the measurement station, a flat V weir with a stable stage-discharge relationship. Even a 0 % degree of tolerance shows no discontinuities within the entire data period. Figure 2b shows the available stage-discharge measurements at Colsterworth.

### 3.2 River Taw at Taw Bridge, UK

In Fig. 3a, a combined BReach$_{t\_2s}$ plot is shown for Taw Bridge. In this plot, time instants near the peak discharges (return period ≥ 2 years) are indicated with a red mark on the bisector. This plot shows changes in data consistency that correspond with historical information. An important change in consistency occurs at index 299 and stage-discharge points after this time






**Figure 3.** (a) Combined BReach$_{t\_2s}$ plot (all data), (b) combined BReach$_{h\_1s}$ plot (data between November 1998 and August 2012) and (c) stage-discharge data for Taw Bridge.





instant are likely to belong to one single consistent period. This starting point corresponds with the moment of installation of a flat V weir (October 1998). Before this date, the data series shows many discontinuities, also for higher degrees of tolerance. The time instants of these discontinuities often coincide with those of the highlighted peak floods. Hence, the plot suggests that these flood events caused geomorphological changes of the river bed that induced changes in consistency and that periods in

between were relatively stable.

The English Environment Agency uses a segmented power law to assess discharges in this measurement station. Rating curve changes generally imply changes of the rating curve coefficients for the lowest and medium flows. The time instants of these official changes are indicated with cyan lines that depart from the bisector. If the change involves also the flood rating curve, an asterisk and (if available) some background information is added to the date indication. Although many of the rating

curve changes correspond with discontinuities, the BReach plot sometimes suggests different or less moments of change.

In Fig. 3b, results of a $BReach_{h\_1s}$ analysis on the stage-discharge data measured after installation of the weir is shown. As can be expected, the plot shows consistency for nearly the complete height range. Only for the highest stages and the lower degrees of tolerance, some discontinuities occur in the plot. Figure 3c shows the available stage-discharge measurements at Taw Bridge. Data measured after installation of the weir are indicated separately.

## 3.3 River Taf at Clog-y-Fran, UK

In Fig. 4a and 5a, combined $BReach_{t\_3s}$ plots based on respectively only winter data and all data are shown for Clog-y-Fran. Although the plot with all data (Fig. 5a) indicates many discontinuities, the maximum reaches of the more tolerant degrees cover a large part of the data set for several points. They are sometimes alternated by data points with more limited reaches. The plot based on only winter data (Fig. 4a) indicates larger consistent blocks. In this latter plot, time instants near the peak

discharges (return period $\geq 5$ years) are indicated with a red mark on the bisector. The time instants of the discontinuities often coincide with those of the highlighted peak floods. Hence, the plots do not only confirm the influence of weed growth, but also suggest that high flood events cause geomorphological changes of the river bed that induce changes in consistency and that periods in between often are relatively stable. Nevertheless, not all floods in Fig. 4a cause discontinuities and not all discontinuities can be linked with the occurrence of large floods. Besides erosion due to large floods, the cross section is also

known to be prone to the (more gradual) build-up of silt. This and other unknown processes might influence the result of this BReach analysis to some extent. Natural Resources Wales, who manage this gauging station, use a segmented power law to assess discharges in this measurement station. In Fig. 4a, the available time instants of these official changes are indicated with cyan lines that depart from the bisector. Although many of the rating curve changes correspond with discontinuities, the BReach plot sometimes suggests different or less moments of change.

Although the flow situation in Clog-y-Fran is complex, the available information about the station can be linked with results of a BReach analysis. These results indicate the need for an in-depth analysis that should lead to an appropriate modeling approach for periods with weed growth. For the remaining (winter) data, an assessment of consistent periods is possible. However, the choice of an appropriate rating curve model is crucial for success. Figures 4b and 4c show results of respectively a $BReach_{t\_1s}$ and a $BReach_{t\_2s}$ analysis on winter data in Clog-y-Fran. The two-segmented rating curve has only a breaking



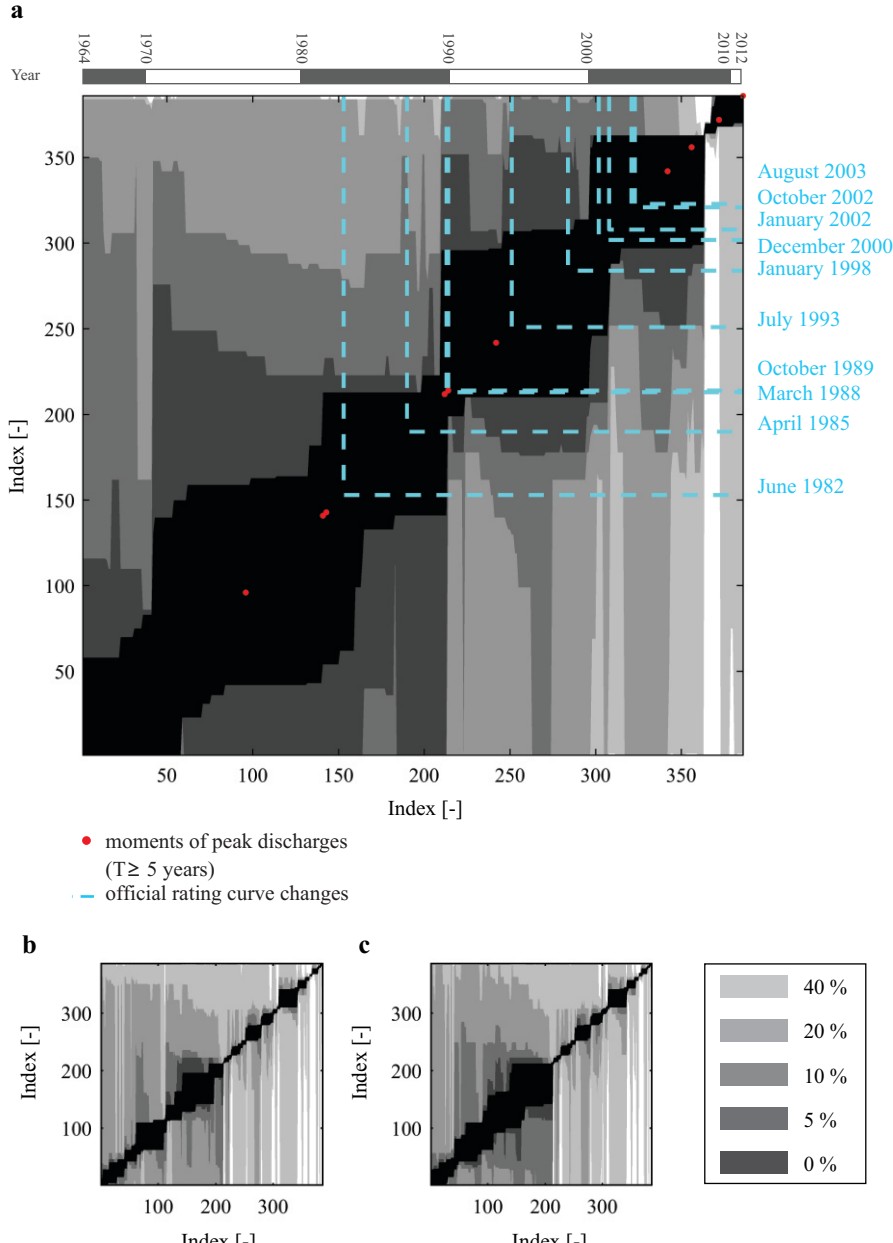

**Figure 4.** Combined (a) BReach$_{t\_3s}$ plot (winter data), (b) BReach$_{t\_1s}$ plot (winter data) and (c) BReach$_{t\_2s}$ plot (winter data) for Clog-y-Fran.

point at the stage of overspill over the right bank. These two figures do not mutually differ a lot, showing that a difference in the rating curve model that affects higher flows has a minor effect on eventual BReach results. This corresponds with earlier





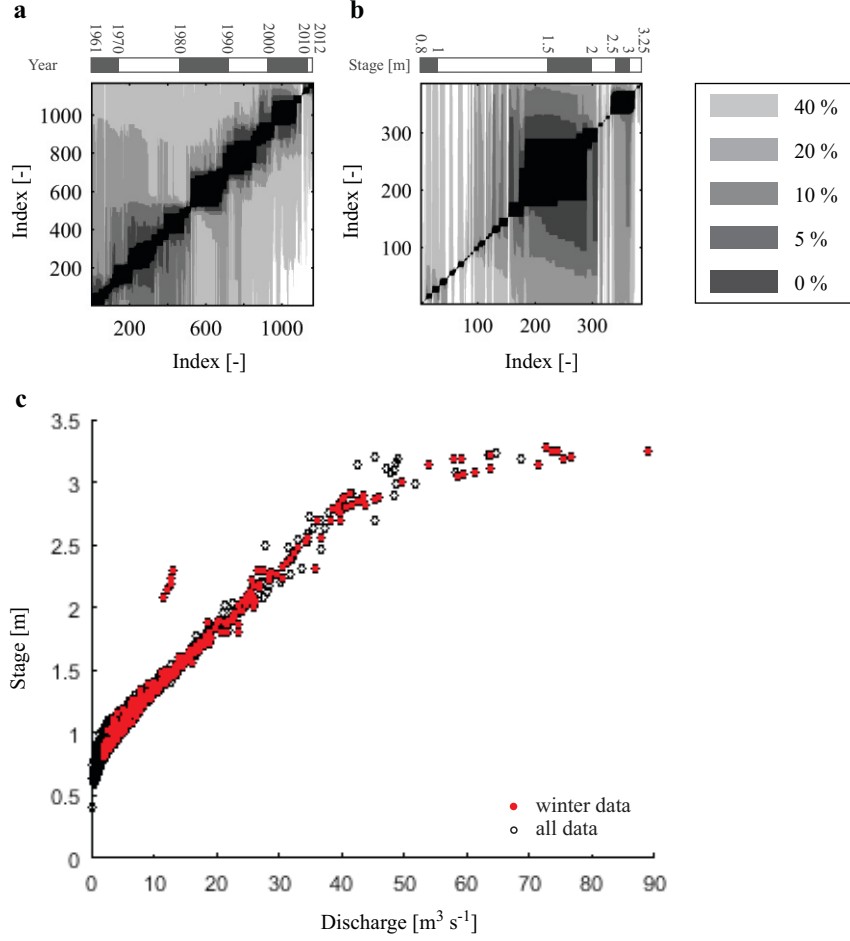

**Figure 5.** (a) Combined $BReach_{t\_3s}$ plot (all data), (b) combined $BReach_{h\_1s}$ plot (winter data) and (c) stage-discharge data for Clog-y-Fran.

results of Van Eerdenbrugh et al. (2016) based on synthetic data. However, comparison of Fig. 4b-4c with Fig. 4a reveals that BReach results alter importantly when adding an extra segment (with a breaking point at low stages) to the rating curve model. In all other stations where a segmented rating curve (two segments) are used, there is a more limited or even a negligible difference with BReach results resulting from a simple power law. Therefore, it is plausible to assume that in Clog-y-Fran, 

5   the flow situation changes from a locally controlled flow (e.g., caused by a riffle affecting the lowest flows) towards a flow situation controlled by a longer river reach for higher flows. It is plausible that this importance of an appropriate modeling of low flow stage-discharge relations on BReach results corresponds with a higher distinctive capacity of these data toward temporal consistency.

    In Fig. 5b, results of a $BReach_{h\_1s}$ analysis of the winter data is shown. The plot shows a relatively large consistency for 

10   stages above index 172 (1.33 m). This is linked with the influence of geomorphological changes on river stages, that is expected



to decrease for increasing discharges due to the corresponding increase of the conveyance of the river cross sections. For high discharges, the order of magnitude of these influences will not exceed the width of the observational uncertainty boundaries anymore and will thus result in more consistent BReach results. Figure 5c shows the available stage-discharge measurements at Clog-y-Fran. Winter data are indicated separately.

5    In the stage-discharge data of Clog-y-Fran (Fig. 5c), four gaugings have discharge values that are smaller than 50 % of all other available discharge measurements with a similar stage. These four observations are all measured on the same day and there is no indication of similar deviations in the months before and after this date. Although it is plausible that these deviations are caused by an erroneous registration of the discharge, there was not enough information to consider these gaugings as outliers. These data occur near the end of the time series (January 2007) and have only a minor effect on the BReach results.

## 3.4    River Pohangina at Mais, New Zealand

In Fig. 6a, a combined BReach$_{t\_2s}$ plot based on measurement uncertainties as applied in McMillan et al. (2010) is shown for Mais. In this plot, time instants near the highest measured stages (return period $\geq$ 1 year) are indicated with a red mark on the bisector. Throughout the whole data set, many discontinuities occur in the plot. The time instants of these discontinuities often coincide with those of the highlighted peak floods. Hence, this plot confirms that in this gravel-bed river, most of these flood events cause geomorphological changes of the river bed that induce changes in consistency and that periods in between are relatively stable.

The Horizons Regional Council interpolates rating curves based on stage-discharge measurements. As these interpolations are changed up to a few times a year, it is not informative to plot these official rating curve changes on the BReach plot.

Figure 6b is a combined BReach$_{t\_2s}$ plot based on measurement uncertainties described by Coxon et al. (2015). There is a high resemblance with Fig. 6a and general conclusions are identical. There are a few reasons for this high resemblance. First, Van Eerdenbrugh et al. (2016) show that a limited misjudgment of observational errors does not alter the conclusions of a BReach analysis fundamentally. Moreover, the classification of results of a rating curve model as acceptable or nonacceptable is based on the assessed uncertainties on both stage and discharge measurements (cfr. Sect. 2.2.3). As mentioned in Sect. 2.4.2, uncertainty boundaries for discharge measurements in Coxon et al. (2015) are substantially larger than in McMillan et al. (2010) while stage uncertainty boundaries are smaller. These opposite differences average the final results. Figure 6c shows the available stage-discharge measurements at Mais.

## 3.5    River Wairau at Barnett's Bank, New Zealand

In Fig. 7a and 7b, combined BReach$_{t\_2s}$ plots are shown for Barnett's Bank, with measurement uncertainties of McMillan et al. (2010) and of Coxon et al. (2015), respectively. Again, both plots are very similar and general conclusions are identical. Figure 7c shows the available stage-discharge measurements at Barnett's Bank.

In Fig. 7a, time instants near the highest measured stages (return period $\geq$ 0.5 year) are indicated with a red mark on the bisector. McMillan et al. (2010) suggest a 0.5 year return period as a threshold that induces consistency changes in this gravel-bed river. This is partly confirmed in the BReach plot, in which discontinuities often (but not always) coincide with the





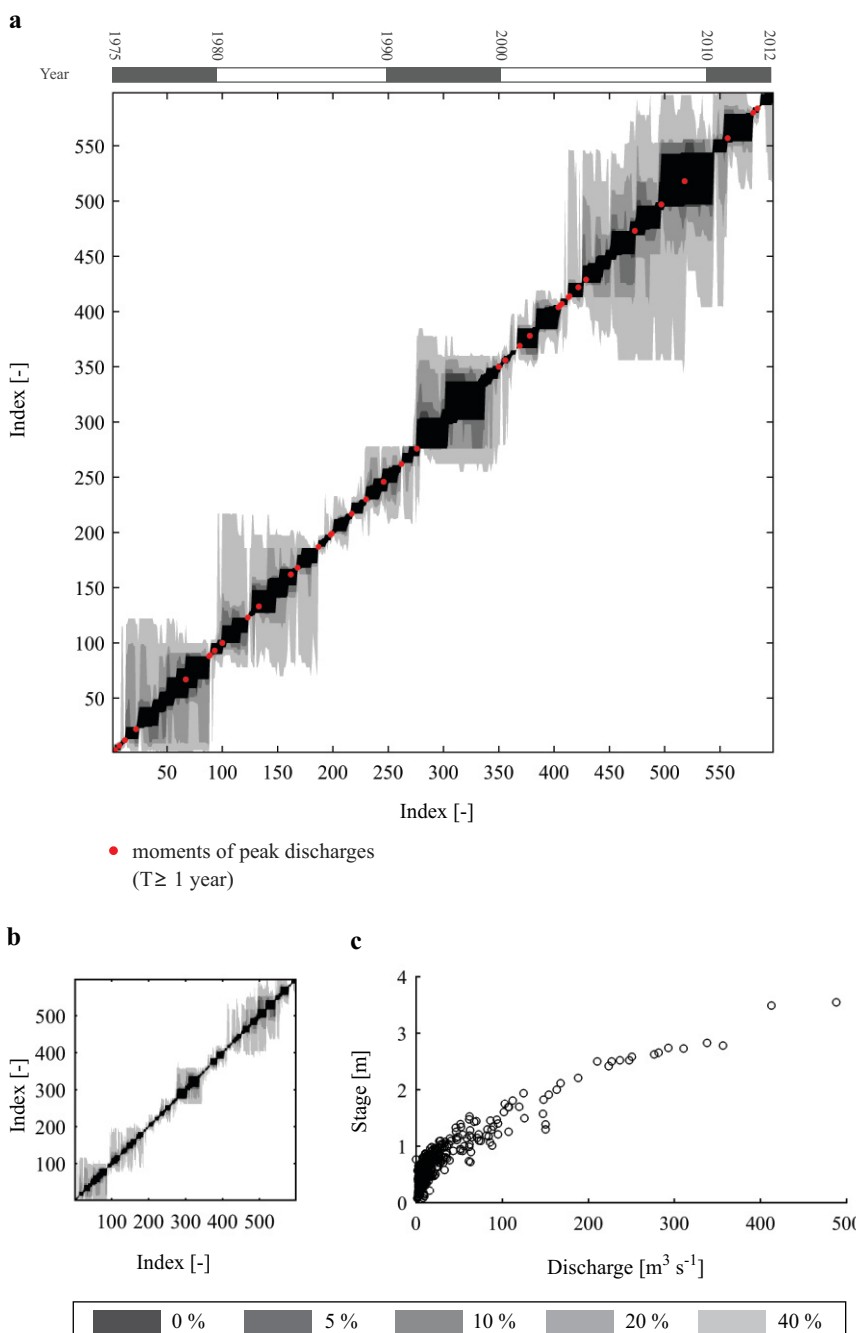

**Figure 6.** Combined BReach$_{t\_2s}$ plot (all data) using uncertainty boundaries from (a) McMillan et al. (2010), (b) Coxon et al. (2015) and (c) stage-discharge data for Mais.





highlighted peak floods that cause geomorphological changes of the river bed. Periods in between these consistency changes are relatively stable. The Marlborough Regional Council interpolates rating curves based on stage-discharge measurements. As these interpolations are changed up to a few times a year, it is not informative to plot these official rating curve changes on the BReach plot.

### 3.6 River Demer at Aarschot, Zichem and Diest, Belgium

In Van Eerdenbrugh et al. (2016), stage-discharge measurements in the river Demer at Aarschot are used to validate the BReach methodology. Although in the current research, a different rating curve model is used and observational uncertainties are assessed slightly different (cfr. Sect. 2.2.1 and 2.4.3), the resulting BReach$_{t\_2s}$ plot (Fig. 8) shows similar results and indicates no consistency before index 29 (August 1982) due to a deepening and widening of the river's cross section and a heightening of the river dikes. After that time instant, a more consistent period starts that lasts until index 233 (February 2005). During the last decade, the data show again a lack of consistency. This is possibly a joint effect of the occurrence of large floods, the introduction of new measurement devices and local maintenance works that affect the cross section of the river bed (Van Eerdenbrugh et al., 2016). In this figure, time instants near the highest measured stages (return period $\geq$ 5 year) are indicated with a red mark on the bisector.

As data are available in two other measurement stations on the river Demer, a comparison between results of these station is interesting. Figure 9a shows combined BReach$_{t\_2s}$ results at Zichem, situated at 16 km upstream of Aarschot. Moments near the highest measured stages (return period $\geq$ 5 year) are indicated with a red mark on the bisector. In Zichem, an important change in consistency is shown at index 72 (December 1988). This corresponds with historical information. In 1988, the river bed near Zichem was deepened and widened and the dikes were heightened, causing the detected consistency change. Before that time instant, the plot shows several discontinuities that possibly suggest changes in consistency. Unfortunately, it was not possible to verify these changes due to a lack of information about this time period. After 1988, the plot suggests the start of a new consistent period until index 144 (March 2008). However, it is difficult to pinpoint the end of this second consistent period precisely. In Zichem, the stage-discharge measurements of March 2008 are the first available measurements since October 2002 and thus this change may already be situated within this period. Since then, the stage-discharge data show nearly no consistency. Again, it is likely that this is a joined effect of several different causes (occurrence of floods, change of measurement device, deviation of the mouth of a small tributary at the location of the measurement station, occasional observations of weed growth in the river).

Figure 9b shows combined BReach$_{t\_2s}$ results in Diest (5 km upstream of Zichem). Moments near the highest measured stages (return period $\geq$ 5 year) are indicated with a red mark on the bisector. In this station, only 34 stage-discharge gaugings measured during one decade are available. Nevertheless, a similar tendency as in the recent data of Aarschot and Zichem can be noticed in the plot. The data are consistent until index 24 (March 2008). Again, it is plausible that this consistency change is linked with the occurrence of peak discharges, with a change in measurement device and with the occasional occurrence of weed in the river bed.



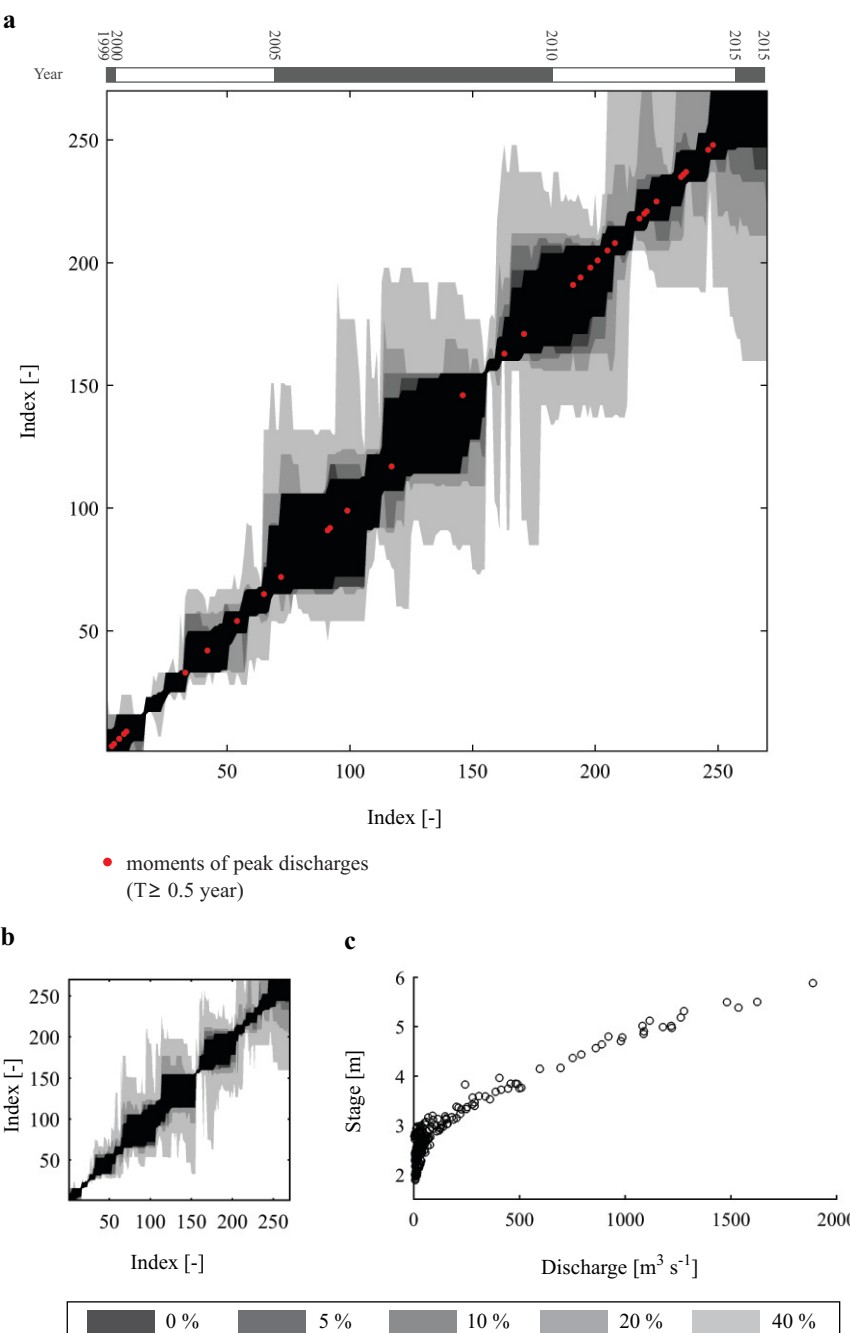

**Figure 7.** Combined BReach$_{t\_2s}$ plot (all data) using uncertainty boundaries from (a) McMillan et al. (2010), (b) Coxon et al. (2015) and (c) stage-discharge data for Barnett's Bank.





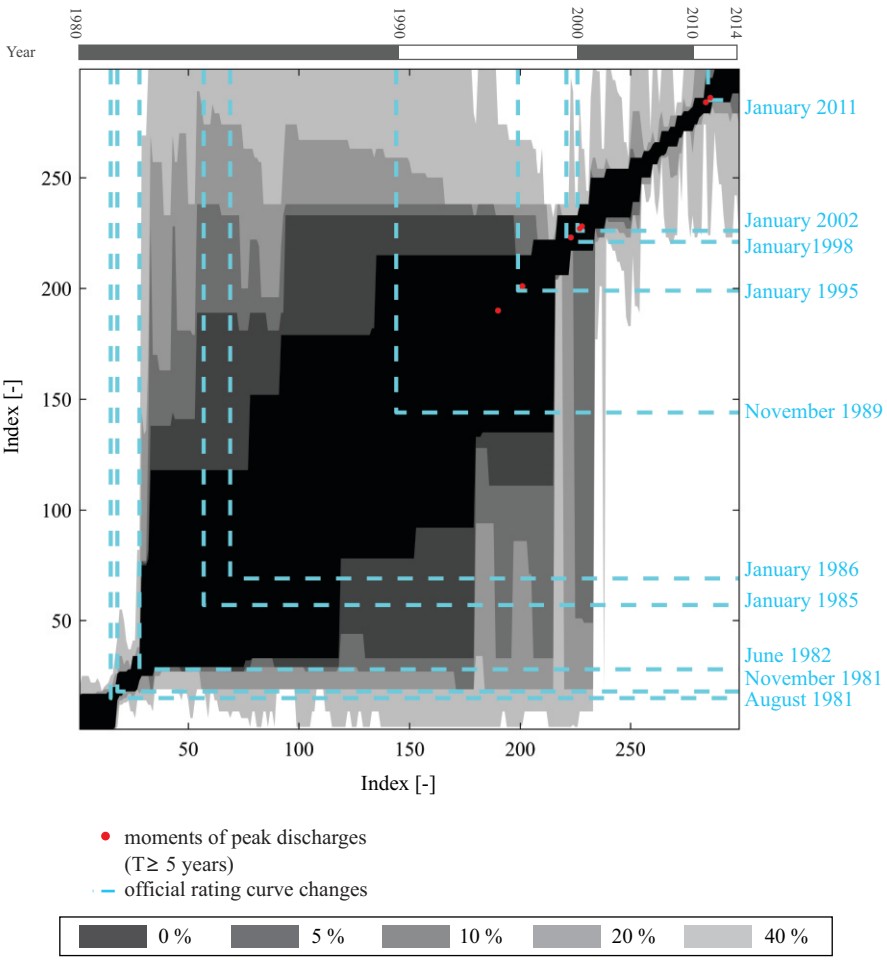

**Figure 8.** Combined BReach$_{t\_2s}$ plot (all data) for (a) Aarschot.

The Flemish Hydrological Information Centre uses a segmented power law to assess discharges in this measurement station. In Fig. 8 and 9a-9b, the time instants of these official changes of the rating curves are indicated with cyan lines that depart from the bisector. Many of the rating curve changes correspond with discontinuities or with the start of a year with a major flood. Nevertheless, the BReach plot sometimes suggests different moments of change. In Fig. 10a-10c, a plot of the available stage-discharge measurements are given for Aarschot, Zichem and Diest. These plots show that for low stages in Aarschot, recently measured discharges (black) are higher than discharges during the long consistent period (red). In Zichem and Diest however, recent discharges tend to be smaller. This latter effect is possibly caused by the observed weed growth in these two stations.



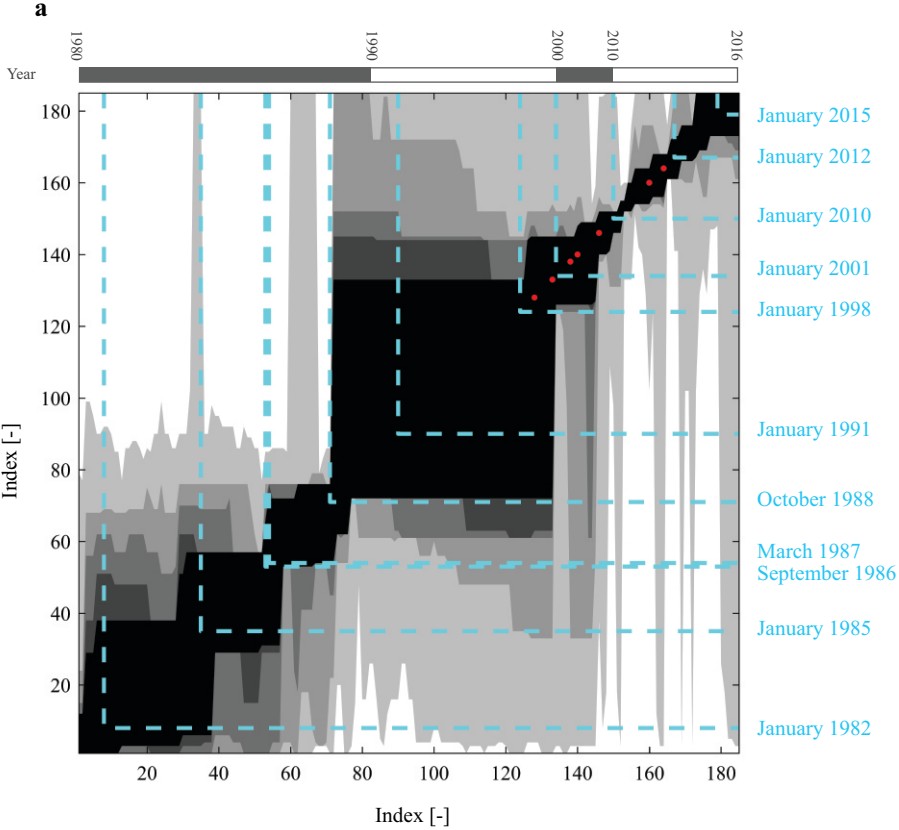

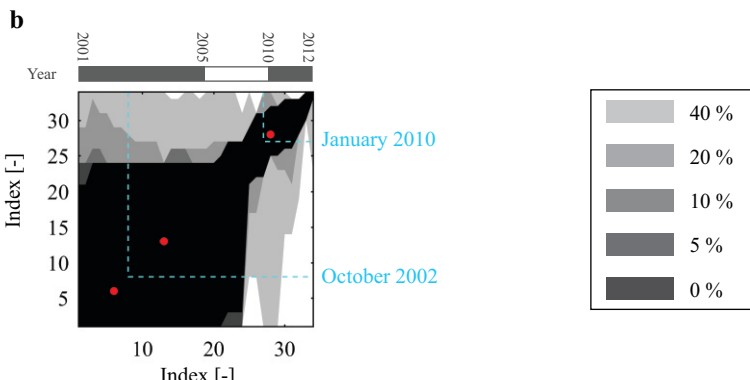

**Figure 9.** Combined BReach$_{t\_2s}$ plot (all data) for (a) Zichem and (b) Diest.





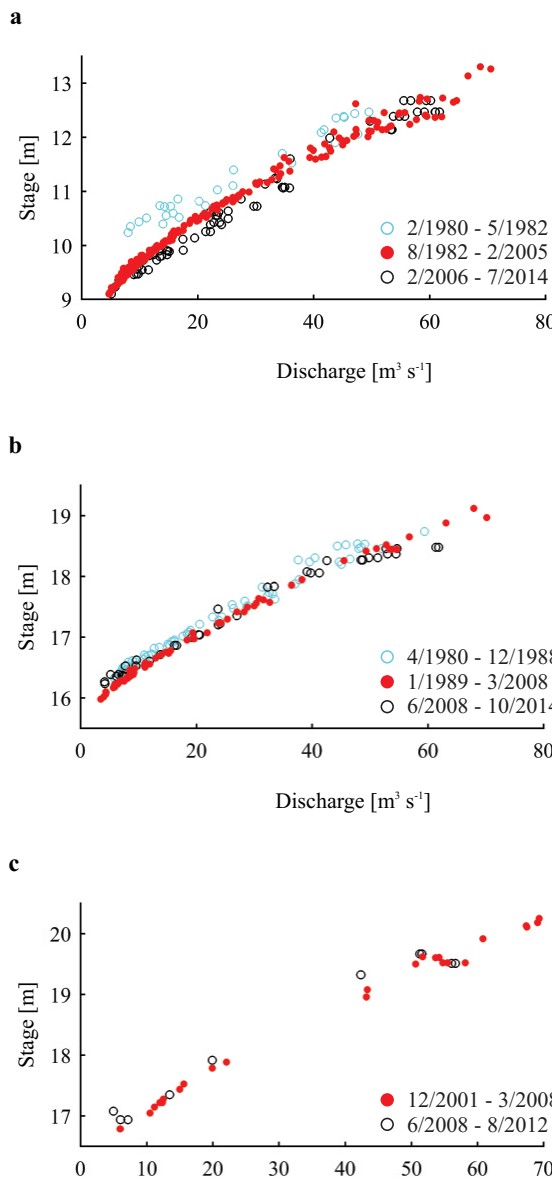

**Figure 10.** Stage-discharge data for (a) Aarschot, (b) Zichem and (c) Diest.

## 3.7 River Grote Nete at Hulshout, Belgium

Figure 11a shows a combined BReach$_{t\_2s}$ plot for Hulshout. Although the plot indicates no consistent periods, the maximum reaches of the most tolerant degree cover almost the complete data set for several data points. They are alternated by data points




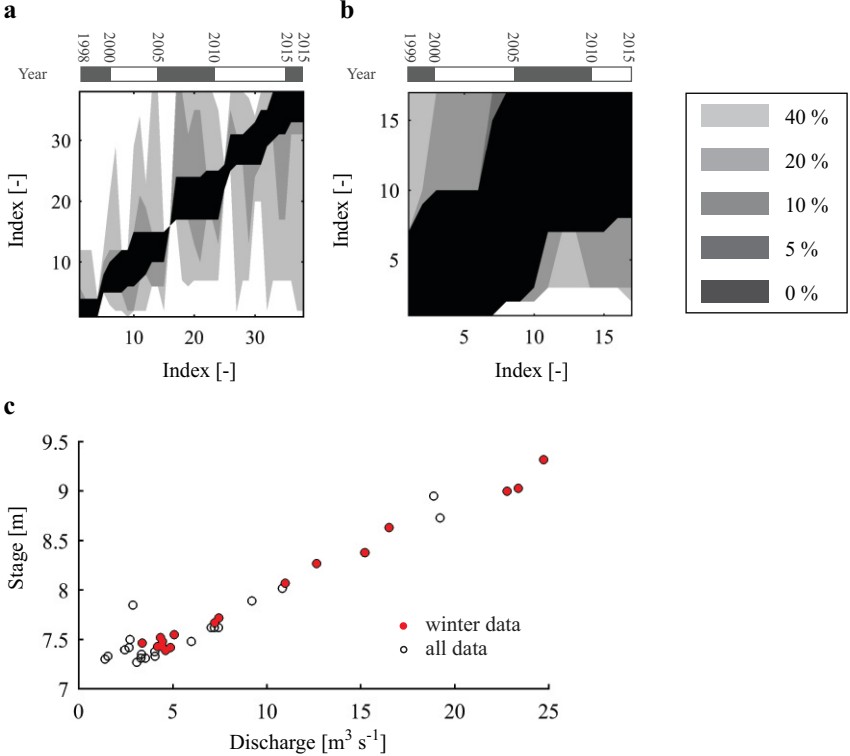

**Figure 11.** Combined BReach$_{t\_2s}$ plot with (a) all data and (b) only winter data, (c) stage-discharge data for Hulshout.

with very limited reaches. Figure 11b shows a combined BReach$_{t\_2s}$ plot of the winter data in Hulshout. For this subset of data, the plot indicates a high consistency for almost the complete period. These results indicate the influence of weed growth and the need for an in-depth analysis that should lead to an appropriate modeling approach for periods with weed growth. Figure 11c shows the available stage-discharge measurements at Hulshout. Winter data are indicated separately.

5    Although the data set is limited to only 38 points, BReach results offer insight in the situation of the measurement station. However, it is likely that a more elaborate data set will result in more robust conclusions.

## 3.8    River Meuse at Maaseik, Belgium

In Maaseik, BReach$_{t\_2s}$ plots of all data points with high degrees of tolerance (Fig. 12a) show an alternation of data points with nearly no reach and data points that have maximum reaches that cover a large part of the data set. In Fig. 12b, results

10    of a BReach$_{h\_1s}$ analysis on the same stage-discharge data are shown. This plot shows no consistency for the lower stages, but indicates a relatively high consistency for stages beyond index 31 (23.46 m). This corresponds with the local situation in Maaseik. Stage-discharge measurements at lower stages are influenced by the downstream movable weir in Linne. For higher





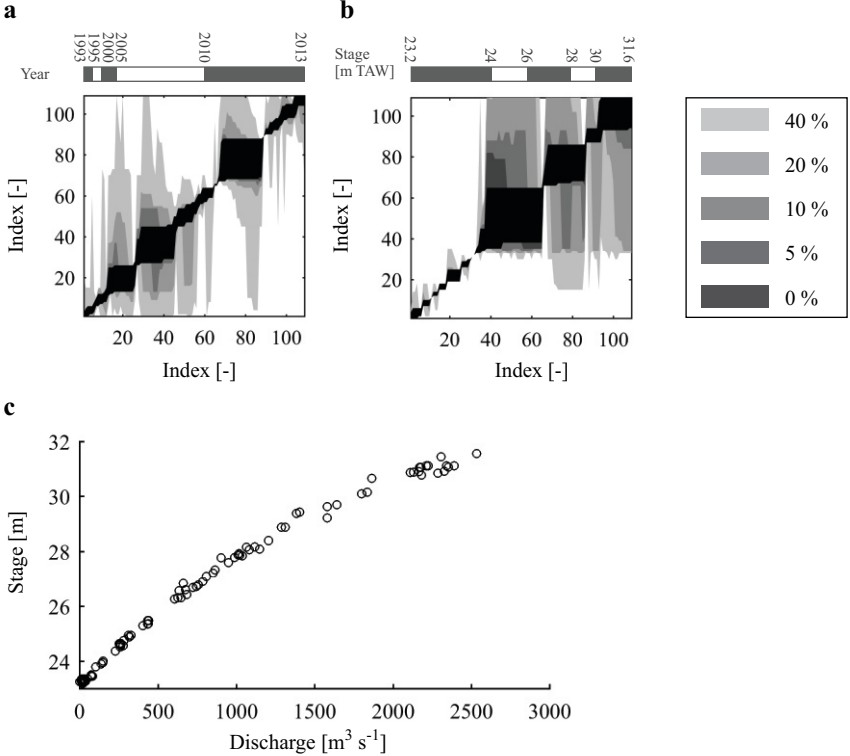

**Figure 12.** (a) Combined BReach$_{t\_2s}$ plot (all data), (b) BReach$_{h\_1s}$ plot (all data) and (c) stage-discharge data for Maaseik.

stages, this influence is smaller. Moreover, the effect of dredging the river bed and of the installed guiding dam on the occurring stages decreases for increasing discharges due to the corresponding increase of the conveyance of the river cross sections. For high discharges, the order of magnitude of these influences will not exceed the width of the observational uncertainties boundaries anymore and will thus result in more consistent BReach results. An in-depth analysis should lead to an appropriate

5    modeling approach for low flow data. Figure 12c shows the available stage-discharge measurements at Maaseik.

### 3.9   General considerations regarding the use of the BReach methodology

In this section, some general thoughts about the use of the BReach methodology for rating curve data are given. It is obvious that the quality of results is related with the gauging frequency of the stage-discharge data. The stations analyzed in this paper vary from densely measured (up to a mean amount of 22 gaugings a year) to rather poorly measured (2 gaugings a year).

10   Stations with a more complex flow situation are measured more frequently. In many cases, local hydrological services decide to apply a similar differentiation in the gauging frequency that depends on the station's complexity. Based on the available data, it was possible to recognize the history and characteristics of each analyzed station in the BReach results. Nevertheless, it





is difficult to pinpoint a minimum required gauging frequency to guarantee a successful application. If a large time gap occurs in the measured data, this can introduce uncertainty about the exact moment of a consistency change. In extreme situations, a temporary change can even disappear from the data resulting in a (misleading) apparently consistent period. The bar (with indication of the years) above a BReach plots permits to detect these noninformative periods. If more detail is wanted, it can be

interesting to create an additional BReach plot in which the absolute time is used in both axes (and thus the indices used in the current plots are projected on these time axes) and with an indication of the moments of the available gaugings on the bisector.

In any case, it is necessary to be informed about the specific situation of the analyzed rating curve station. Not only is it important for an adequate choice of a rating curve model (cfr. Sect. 3.3), it is as well required for a correct interpretation of the BReach results and the design of possible alternative BReach analyses (Sect. 2.3). For instance, it would not be possible

to distinguish between the BReach$_t$ results of all available data at Hulshout and Maaseik (Sect. 3.7 and 3.8) without any knowledge of the local situation.

The computational load of the BReach methodology depends on several aspects. First, it increases linearly with the size of the sample of the parameter space (and is thus larger for more complex rating curves). Second (and more important), the necessary calculation time strongly depends on both the amount of stage-discharge data points and the degree of consistency

of the data set. The principle of the BReach algorithm is that for each data point, a maximum left and right reach must be searched. If a data set is highly consistent, the length of these searches increases significantly. Doubling the amount of data points can (for consistent data sets) hence result in eight times the original calculation time. In the research for this paper, all calculations are performed on a personal computer with a 3.4 GHz CPU Core I7 and 8 GB RAM. For most stations, a BReach analysis took a few minutes to a few hours. In the most complex case (Clog-y-Fran, with 1166 data points and $1.3 \times 10^7$

samples), calculation of BReach results required 72 hours.

At the moment, interpretation of BReach results are done manually by the user. The availability of a (semi-)automatic routine that identifies possible consistent data periods would improve the BReach methodology. As the degree of squareness of a BReach plot within a certain period expresses the lack of important discontinuities, it might play a role in the decision process for assessing consistent periods.

**4  Conclusions**

In this paper, the BReach methodology to assess temporal consistency in rating curve data is successfully tested on various stage-discharge data set in the UK, New Zealand and Belgium. For each country, local information is maximally used to estimate observational uncertainties that serve as an input for the methodology. In this context, a new approach is proposed for the Belgian data using relative differences between simultaneous discharge measurements to test the plausibility of several

a priori assumed error distributions. This approach offers promising insights in the plausible character of measurement error distributions in addition to a more general use of existing literature data about observational uncertainties. However, the limited size of the data set with simultaneous measurements is an important restriction. In order to investigate the possibilities of the proposed approach more profoundly, a more elaborate data set with large spread in time, measurement stations, measurement





device and flow conditions is necessary. Such an enlarged data set would not only increase the reliability of a KS test, but would also enhance the possibility to use more bins with smaller ranges of normalized discharge (replacing the current two arbitrary subgroups) and to investigate other measurement devices.

Overall, results of the BReach analyses correspond with site-specific situations. Nevertheless, the investigated cases show
that knowledge about the local situation of a measurement station is crucial to design the necessary BReach analyses and to interpret their results correctly. Results show consistency in locations that are known as stable. Where human interventions (e.g., installation of a weir, deepening of a river) altered the rating curve behavior, results show corresponding consistency changes. In locations influenced by weed growth, a higher consistency can be assessed after isolating winter data. Similarly, consistency can be assessed for higher stages in a station where a downstream weir influences low flow behavior. Stations that
are prone to geomorphological changes caused by flood events show discontinuities in the BReach plots at the time instants of the highest floods. Moreover, the plots can also indicate which peak floods do not cause consistency changes. The return period that serves as a threshold for consistency changes varies from station to station. These results provide extra insight into the rating curve behavior and confirm the added value of the proposed BReach methodology as a preliminary assessment of data consistency prior to an in-depth determination of discharges and their uncertainty. Moreover, this assessment of (in)consistent
periods can enhance other applications based on the investigated data (e.g., by informing hydrological and hydraulic model evaluation design about consistent time periods to analyze).

In the BReach methodology, the chosen rating curve model is required to appropriately approximate the relation between discharge and stage for an important part of the measured range. In this paper, analyses with only a subset of the data or with stage-discharge data sorted by stage ($\text{BReach}_h$) enable to overcome a part of a known model deficiency.

Results of Van Eerdenbrugh et al. (2016) show that stage-discharge data for higher stages have a smaller distinctive capacity in the $\text{BReach}_t$ analysis. This corresponds with results of Di Baldassarre and Claps (2011), who confirm the validity of one single flood rating curve throughout a period with different geometric situations (affecting the rating curve for lower flows). A limited deficiency of the rating curve model for the heighest flows leads to satisfying BReach results as the effects of the model deficiency disappear from the plots with higher degrees of tolerance (Van Eerdenbrugh et al., 2016). In the current paper, results
in Clog-y-Fran (Sect. 3.3) confirm these findings. It is however important to emphasize that these results are site-specific and are expected to depend on the extent to which the higher parts of the cross section contribute to changes in the flow situation. On the contrary, a model deficiency in a height range that contributes significantly to changes in the flow situation will lead to important changes in $\text{BReach}_t$ results. This is shown in this paper for low stages at Clog-y-Fran (Sect. 3.3). When applying the BReach methodology, it is thus advisable to select a best possible model structure based on the available knowledge about
flow conditions in the investigated measurement site.

## 5   Code availability

The BReach code is available from the authors upon request (katrien.vaneerdenbrugh@ugent.be).



## 6 Data availability

The New Zealand rating curve data were obtained freely from Horizons Regional Council (http://www.horizons.govt.nz/contact-us) and from Marlborough District Council (mdc@marlborough.govt.nz). The UK rating curve data were obtained freely from Environment Agency (enquiries@environment-agency.gov.uk) and Natural Resources Wales (enquiries@naturalresourceswales.gov.uk).

5   The Belgian rating curve data were obtained freely from the Flemish Hydrological Information Centre (hic@vlaanderen.be).

*Author contributions.*   K. Van Eerdenbrugh, S. Van Hoey and N. Verhoest developed the BReach methodology and designed the overall setup of the analyses. G. Coxon and J. Freer provided observational uncertainties for the UK data and contributed to the design of the analyses and the interpretation of results for these data. K. Van Eerdenbrugh assessed observational uncertainties for Belgian data, performed the BReach analyses and prepared the manuscript with contributions from all co-authors.

10   *Competing interests.*   The authors declare that they have no conflict of interests.

*Acknowledgements.*   The authors thank Flanders Hydraulics Research and its Hydrological Information Centre for providing gauging data and accompanying information and Waterwegen en Zeekanaal NV for providing information about infrastructure works in Aarschot, Zichem and Diest. Furthermore, they thank Brent Watson for providing additional information about the measurement station in Mais. This research has benefitted from a statistical consult with Ghent University FIRE (Fostering Innovative Research based on Evidence). Gemma Coxon and

15   Jim Freer were supported by NERC MaRIUS: Managing the Risks, Impacts and Uncertainties of droughts and water Scarcity, grant number NE/L010399/1.





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

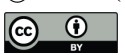



Table 1: Study Sites and their Characteristics

| Station name | River | Country | Upstream catchment [km²] | Mean flow [m³ s⁻¹] | First h-Q measurement | Last h-Q measurement | $h_{min,cont}$ [m] | $h_{max,h-Q}$ [m] | Amount of h-Q measurements | Characteristics related with rating curve behavior and data consistency |
|---|---|---|---|---|---|---|---|---|---|---|
| Colsterworth | Witham | UK | 51.3 | 0.23 | 1984 | 2012 | 0.08 | 0.78 | 99 | Flat V-weir, stable rating curve. |
| Taw Bridge | Taw | UK | 71.4 | 1.85 | 1968 | 2012 | 0 | 1.15 | 412 | Original situation: unstable bed control. Stable rating curve since installation of a Flat V-weir (September-November 1998). |
| Clog-y-Fran | Taf | UK | 217.3 | 7.6 | 1961 | 2012 | 0 | 0.16 | 1166 | Considerable overspill over the right bank at a stage of approximately 3.2-3.4 m. Weed growth affecting low flow behavior. Channel prone to build-up of silt. |
| Mais | Pohangina | New Zealand | 488 | 19$^a$ | 1975 | 2014 | 0.13 | 1.13 | 598 | Unstable (gravel) bed control. |
| Barnett's Bank | Wairau | New Zealand | 3825 | 100$^b$ | 1999 | 2015 | 0 | 0.98 | 270 | Unstable (gravel) bed control. |
| Aarschot | Demer | Belgium | 2146 | 14.3 | 1980 | 2014 | 0.14 | 0.02 | 299 | Small backwater effect at high flows. Important deepening and widening of the river bed until July 1982. Introduction of new measurement devices since February 2006. Maintenance works affecting the river bed between May 2007 and December 2010. |

$h_{min,h-Q}$ ($h_{max,h-Q}$): minimum (maximum) stage value of all available stage-discharge measurements
$h_{min,cont}$ ($h_{max,cont}$): minimum (maximum) value of all available continuous stage measurements
$^a$ Ibbitt and Pearson (1987)
$^b$ Wilson and Wöhling (2015)





| Station name | River | Country | Upstream catchment [km²] | Mean flow [m³ s⁻¹] | First h-Q measurement | Last h-Q measurement | $h_{min,h-Q}$ ($h_{min,cont}$) [m] | $h_{max,h-Q}$ ($h_{max,cont}$) [m] | Amount of h-Q measurements | Characteristics related with rating curve behavior and data consistency |
|---|---|---|---|---|---|---|---|---|---|---|
| Zichem | Demer | Belgium | | 13.4 | 1980 | 2016 | 0.03 | 0.02 | 185 | Important deepening and widening of the river bed in 1988. Completion of deviation of the mouth of a small tributary at the location of the measurement station in 2003. Introduction of new measurement devices since March 2008. Occasional weed growth (registered since 2011). |
| Diest | Demer | Belgium | | 13.3 | 2001 | 2012 | 0.43 | 0.01 | 34 | Introduction of new measurement devices since March 2008. Occasional weed growth (registered since 2011). |
| Hulshout | Grote Nete | Belgium | 443.5 | 4.83 | 1998 | 2015 | 0.16 | 0.13 | 38 | Weed growth affecting low flow behavior. |
| Maaseik | Meuse | Belgium | 21787 | 249.5 | 1993 | 2013 | 0.07 | 0.26 | 109 | Downstream movable weir affecting low flow behavior. Dredging of the winter bed and change of the local flow situation under the bridge at the measurement station in 2008. |

$h_{min,h-Q}$ ($h_{max,h-Q}$): minimum (maximum) stage value of all available stage-discharge measurements
$h_{min,cont}$ ($h_{max,cont}$): minimum (maximum) value of all available continuous stage measurements




**Table 2.** Kolmogorov-Smirnov test results (simultaneous discharge measurements)

| Data set | Error distribution type | Minimum KS statistic [-] | Maximum adjusted p-value [-] | Scale parameter [%] | 95 % uncertainty boundaries [%] |
|---|---|---|---|---|---|
| Low flow data | Gaussian distribution | 0.14 | 0.79 | 3.12 | ± 6.12 |
| Low flow data | Logistic distribution | 0.14 | 0.79 | 1.80 | ± 6.59 |
| High flow data | Gaussian distribution | 0.13 | 0.82 | 1.90 | ± 3.72 |
| High flow data | Logistic distribution | 0.13 | 0.81 | 1.10 | ± 4.02 |

Low flow data have values for $Q_n$ between 0.72 and 3.64 and high flow data have values for $Q_n$ between 3.72 and 8.41

**Table 3.** Values for $h_{br,max}$

| Station | $h_{br,max}$ [m] |
|---|---|
| Taw Bridge | 0.4 |
| Mais | 1 |
| Barnett's Bank | 3 |
| Aarschot | 10 |
| Zichem | 16.4 |
| Diest | 17.2 |
| Hulshout | 7.5 |
| Maaseik | 23.4 |