# Peer review of "Consistency assessment of rating curve data in various locations using Bidirectional Reach (BReach)"

_Hydrology and Earth System Sciences, 2017_

## Referee Comment (RC1) · Anonymous Referee #1 · 18 Jun 2017

Detecting instable stage – discharge relationships, and the generation of rating curves under such conditions, are considered as major problems in hydrometry. Methods for general cases do not exist in the literature. The present paper considers a general approach to the detection of instable/stable periods. It is however disappointing to see that the generation of rating curves under instability is not considered, but this does not hinder the paper from being interesting. The paper is well written and the case studies are comprehensive. Plots and figures are fine. The basic method used stems from another published paper, and its technical characteristics are therefore of less importance here. All in all, the material should be interesting to read for hydrologists and for hydrographers in particular. But in my opinion the study must be slightly improved

before it is ready for publication.

I do not feel convinced about the capabilities of the BReach technique after having read the paper. It assesses consistency based on a fixed rating curve model and a fixed sampling space for parameter values assumed plausible. General, or average, values on measurement uncertainty from various sources are applied to justify the acceptable zone for measurements. Besides these intrinsic limitations, the method is not compared to a simpler and established method to assess any novel capabilities.

One has to select the number of segments and associated break-points before the analysis in the BReach method. This procedure probably introduces at least two problems. First, assume that there are no channel changes, but that the segmentation model used is inappropriate. Some ranges will be affected more than others of this model error. Can this lead to problems (i.e. consecutive measurements in such areas can lead to the BReach method to indicate so-called discontinuities) and if so, what can be done to avoid them? The authors should provide an answer to this in the form of a discussion in a section prior to the application of the method. Second, the values in the assessment of the uncertainties on measured stage and discharge are based on material where the correct segmentation model is assumed. The tolerance limits applied are also built on the presupposition of a correct segmentation model in the simulation. Can this lead to problematic results and if so, can anything be done to minimize the effect? The authors should provide answers in the same section as suggested above.

The application of the BReach method is rather comprehensive in the study. Many case studies are used. It can be debated on how accurately the results fit with the prior information on channel changing characteristics. To convince me about the appropriateness of the BReach framework, a simpler and established method must also be applied to the case studies. More precisely, a rating curve model with similar segmentation characteristics could have been applied to all measurements. A simple analysis of the corresponding residuals (residual – time plots) can then act as a fair comparison.

---

## Referee Comment (RC2) · Anonymous Referee #1 · 19 Jun 2017

Rating curve fitting is basically a nonlinear regression problem. Traditionally, post-modelling analysis in regression is based on residual investigations. Auto-correlated residuals (non-stationarity) are, basically, what one is dealing with in cases of channel change. Hence, classical residual analysis is the common tool available for assessing stationarity in the stage – discharge relationship. A novel approach such as the BReach method should therefore be compared with such a common, simple and, more important, independent method in order to illustrate the strength and appropriateness of the new technique proposed. It is possible that other known objective methods exist in hydrometry for detecting stable and instable rating periods. If the authors believe that there are no comparable methods that are worth using for comparison, they should

at least give some explanation and evidences for this in the paper. This in itself is an argument for the importance of their method.

---

## Short Comment (SC1) · 19 Jun 2017

Dear reviewer, thank you for the useful reply. Most of your reactions are clear to me and help to provide an added value to the paper. However, I have a question concerning your last remark (comparation with an established method) in order to understand it appropriately. Do I understand correctly that your suggestion is to determine a single rating curve (based on a well-chosen evaluation criterium) for (1) the complete data series of a station and (2) the delineated consistent periods using the BReach methodology in order to compare and - if possible - classify the (chronologically sorted) residuals? Thank you in advance for your reaction. Best regards, Katrien Van Eerdenbrugh

---

## Referee Comment (RC3) · K. Engeland (Referee) · 14 Jul 2017

The paper demonstrates a methodology developed by Van Eerdenburgh (2016) for detecting in-consistency, e.g. step changes or trends, in rating curve models. It deserves publication after some modifications. Please see my comments below.

1: In the introduction it could be useful to explicitly define objectives (and if necessary sub-objectives). It makes the paper easier to read and it will be easier to write more crisp conclusions.

2: The main part of the last paragraph in the conclusions fits better into the discussion

section (section 3.9).

3: In the explanation of the bidirectional reach step 5, I miss a small clarification since it is possible that more than one parameter sets could be regarded as acceptable for a set of data points. I think this sentence from Van Eerdenbrugh (2016) gives a nice explanation: "the vertical distance between the bisector and the maximum left reach indicates the maximum amount of data points before the investigated data point that can be described by at least one parameter set and under the prevailing degree of tolerance".

4: It might be confusing to use the terms "left" and "right" reach. in the plots it is the vertical span of the grey-shades that shows the "left" and "right" reaches. Would it be better to call it "lower" and "upper" reach?

5: I have one question about the shape of the BReach plots. The black shaded areas seems to be well symmetric around the bi-sector, whereas for the other tolerance levels, the symmetry around the bi-sector is lost. Why is it so? I would expect symmetry around the bi-sector. One example: if you at data point 20 look forwards and find that until data point 60, at least one rating curve model falls outside the error bars at less than 5% of the data points, you would get the same result if you look backwards from data point 60 towards point 20? Is the explanation that you have a directional search for the left (and right) reaches, and that the data points where the search stops, depends on from which direction you start the search?

6: Why do you consider the 0% tolerance in the BReach plots? Would it not be more correct to set the minimum tolerance at 5% since you used 95% C.I. to define an interval for the measurement errors?

7: The figures are not well explained by the information given in the legend and the figure captions. The legend with the grey shades could have "Tolerance" or "Tolerance level" as a title. In the caption it would also be nice to add one or two sentences that explain the plot briefly. Something like "The shaded areas below and above the

[Figure]

bisector shows the left and the right reach (vertical axis) of each data point (horizontal axis). The tolerance levels indicate the maximum tolerated ratio of data points for which one (or more) rating curve models are outside the measurement uncertainty"

8: The grey-shades used in the plots do not correspond to the grey-shades in the legend. In particular, the 0% error bar seems to be black in the plot and a grey-shade in the legend.

9: Standard regression analysis also provide tools for analyzing residuals and in particular other methods are available for detecting step-changes in ordered data. It would be interesting to compare your results to existing methods.

---

## Editor Comment (EC1) · B. Schaefli (Editor) · 7 Aug 2017

Both reviewers think that the paper is interesting for the readership of HESS and give detailed indications about how to improve the manuscript. I would like to invite the authors to respond to both reviews as soon as possible before preparing a revised version.

---

## Author Comment (AC1) · 28 Aug 2017

Comments from referees and author's response on "Consistency assessment of rating curve data in various locations using Bidirectional Reach (BReach)" by Katrien Van Eerdenbrugh et al.

*Italic black text: original comments of the reviewers*
Blue text: author's response
Narrow black text: fragments from the article (adapted or added text)

In response to the reviewers helpful comments, substantial changes have been made (including additional analysis comparing the method to a simpler approach and additional text added in the methodology, results and discussion). Details about the changes are described in this document.
* * *
Anonymous Referee #1
* * *
*Detecting instable stage – discharge relationships, and the generation of rating curves under such conditions, are considered as major problems in hydrometry. Methods for general cases do not exist in the literature. The present paper considers a general approach to the detection of instable/stable periods. It is however disappointing to see that the generation of rating curves under instability is not considered, but this does not hinder the paper from being interesting. The paper is well written and the case studies are comprehensive. Plots and figures are fine. The basic method used stems from another published paper, and its technical characteristics are therefore of less importance here. All in all, the material should be interesting to read for hydrologists and for hydrographers in particular. But in my opinion the study must be slightly improved before it is ready for publication.*

*I do not feel convinced about the capabilities of the BReach technique after having read the paper. It assesses consistency based on a fixed rating curve model and a fixed sampling space for parameter values assumed plausible. General, or average, values on measurement uncertainty from various sources are applied to justify the acceptable zone for measurements. Besides these intrinsic limitations, the method is not compared to a simpler and established method to assess any novel capabilities.*

We thank reviewer #1 for these comments and address them in the responses below.

*__1:__ One has to select the number of segments and associated break-points before the analysis in the BReach method. This procedure probably introduces at least two problems. First, assume that there are no channel changes, but that the segmentation model used is inappropriate. Some ranges will be affected more than others of this model error. Can this lead to problems (i.e. consecutive measurements in such areas can lead to the BReach method to indicate so-called discontinuities) and if so, what can be done to avoid them? The authors should provide an answer to this in the form of a discussion in a section prior to the application of the method.*

*Second, the values in the assessment of the uncertainties on measured stage and discharge are based on material where the correct segmentation model is assumed. The tolerance limits applied are also built on the presupposition of a correct segmentation model in the simulation. Can this lead to problematic results and if so, can anything be done to minimize the effect? The authors should provide answers in the same section as suggested above.*

Author's response: A paragraph is dedicated to discuss this remark. It is added at the end of Section 2.2.1 (Step 1: Selection of a model structure for the analysis)
Changes in text: p. 5 lines 11 - 29

Generally, the choice of rating curve model should maximally be based on the existing flow situation at the rating curve station. In case more complex flow situations (e.g., hysteresis or backwater effects) are observed and described, it is possible to apply the BReach methodology with an adapted rating curve model (e.g., Jones, 1916; Petersen-Øverleir, 2006; Dottori et al., 2009; Reitan and Petersen-Øverleir, 2011). In case there is little or no knowledge of the flow situation, it is tempting to use a rating curve model with multiple segments and wide sample ranges for the breakpoints. If the amount of samples is sufficiently large, the possibility of obtaining nearly identical values for the parameters of two adjacent segments theoretically enables to eliminate an excess of segments in the chosen model. As shown in the example at Clog-y-Fran (Sect. 3.3), the parameter sets that result in a model structure with the largest maximum reaches will be decisive for eventual BReach results. This approach however involves the risk of overfitting the model to the available gauging data, mainly in case of small and inconsistent stage-discharge data sets. It is not implausible that in such a case of sparse gauging data, eventual BReach results are obtained by a model structure that is not capable to describe the real flow situation at the site, but instead incidentally fits a series of consecutive gauging points that not only belong to different height ranges but also to different consistent periods. Therefore, and similarly as in other rating curve applications, the choice of an appropriate rating curve model should preferably be based on a hydraulic analysis of the measurement site (Le Coz et al., 2014).

It is important to mention that all decisions to be made in the BReach methodology, such as the assessment of the measurement uncertainty (Sect. 2.2.3 and 2.4) and of different degrees of tolerance (Sect. 2.2.4) are made independently of the appropriateness of the chosen rating curve model. Despite of the methodology's ability to account for a limited model deficiency (Sect. 2.2.4 and Van Eerdenbrugh et al. (2016)), this additionally advocates a well-considered choice of a model structure.

*2:* *The application of the BReach method is rather comprehensive in the study. Many case studies are used. It can be debated on how accurately the results fit with the prior information on channel changing characteristics. To convince me about the appropriateness of the BReach framework, a simpler and established method must also be applied to the case studies. More precisely, a rating curve model with similar segmentation characteristics could have been applied to all measurements. A simple analysis of the corresponding residuals (residual – time plots) can then act as a fair comparison.*

Author's response: A residual analysis based on the set of parameters that minimizes the root mean square error for the complete data set is performed for all stations where assumptions concerning discharge measurement uncertainties allow for this approach (i.e., homoscedastic relative errors that follow a Gaussian distribution). For different groups of stations, this analysis led to similar patterns in the results. Therefore, three representative stations are included in the paper (Maaseik, Aarschot and Barnett's Bank). A visual interpretation of the residual plots is performed and discussed.
Changes in text: A paragraph is added in Sect. 2 (Methods, p. 12, line 26 – p. 13, line 2), Sect. 3 (Results and discussion, p. 21, lines 1-22) and Sect. 4 (Conclusions, p. 23 lines 23 – 24)

[revised manuscript text omitted]

K. Engeland (Referee #2)

*The paper demonstrates a methodology developed by Van Eerdenburgh (2016) for detecting inconsistency, e.g. step changes or trends, in rating curve models. It deserves publication after some modifications. Please see my comments below.*

*__1:__ In the introduction it could be useful to explicitly define objectives (and if necessary sub-objectives). It makes the paper easier to read and it will be easier to write more crisp conclusions.*
Author's response: The last paragraph of the introduction is slightly adapted to formulate the paper's objective (perform an additional analysis with more diverse measured data sets in order to further explore the methodology's applicability) more explicitly. The benchmark using a residual analysis (cfr. Response 2 to reviewer #1 and Response 9 to reviewer #2) is added to the description of the actions that serve this objective. The conclusions start with a reference to the paper's objective.
Changes in text: Introduction (p. 3, lines 5 – 12) and Conclusions (p. 22, lines 31-32)

**1 Introduction**
… All investigated data sets in this study belong to the same geographical location. Therefore, the objective of the current paper is to perform an additional analysis with more diverse measured data sets in order to further explore the methodology's applicability. For this purpose, several gauging stations in the United Kingdom (UK), New Zealand and Belgium are selected based on their well-documented history and their specific characteristics related to rating curve consistency. For each country, regional information is maximally used to estimate observational uncertainty. Based on this uncertainty, a BReach analysis is performed and subsequently, results are validated against available knowledge about the history and behavior of the site. In a selection of the investigated stations, results of the BReach methodology are additionally compared with results of a classical residual analysis.

**4 Conclusions**
The objective of this paper was to test the BReach methodology to assess temporal consistency in rating curve data on various stage-discharge data set in the UK, New Zealand and Belgium. This led to successful results for all tested sites.
…

*__2:__ The main part of the last paragraph in the conclusions fits better into the discussion section (section 3.9).*
Author's response: part of the paragraph is moved to Section 3.10 (General considerations regarding the use of the BReach methodology, p. 22, lines 3 – 11).
Changes in text: part of the paragraph is moved to Section 3.10 (General considerations regarding the use of the BReach methodology, p. 22, lines 3 – 11).

*__3:__ In the explanation of the bidirectional reach step 5, I miss a small clarification since it is possible that more than one parameter sets could be regarded as acceptable for a set of data points. I think this sentence from Van Eerdenbrugh (2016) gives a nice explanation: "the vertical distance between the bisector and the maximum left reach indicates the maximum amount of data points before the investigated data point that can be described by at least one parameter set and under the prevailing degree of tolerance".*
Author's response: As nearly the same sentence was already included in the explanation of step 5, the purpose of this comment was not clear to the authors. Therefore, the original text is not changed. (Original text: "The vertical distance between the bisector and the index of the maximum left reach represents the maximum amount of data points before the investigated data point that can be described with at least one set of parameters under the chosen degree of tolerance.")
Changes in text: none.

*__4:__ It might be confusing to use the terms "left" and "right" reach. in the plots it is the vertical span of the grey-shades that shows the "left" and "right" reaches. Would it be better to call it "lower" and "upper" reach?*

Author's response: The authors opt not to adapt these terms. The terms "upper" and "lower" reaches can be helpful with regard to the interpretation of the plots. However, for understanding the meaning of these reaches (a temporal span for which a rating curve model behaves satisfactory), a mental visualization of a time series in a vertical direction would be required from the reader, which is rather contra-intuitive.

Changes in text: none.

**5:** *I have one question about the shape of the BReach plots. The black shaded areas seems to be well symmetric around the bi-sector, whereas for the other tolerance levels, the symmetry around the bi-sector is lost. Why is it so? I would expect symmetry around the bisector. One example: if you at data point 20 look forwards and find that until data point 60, at least one rating curve model falls outside the error bars at less than 5% of the data points, you would get the same result if you look backwards from data point 60 towards point 20? Is the explanation that you have a directional search for the left (and right) reaches, and that the data points where the search stops, depends on from which direction you start the search?*

Author's response: Although there is some symmetry in the general shape of the figures, the plots with non-zero tolerance degrees are indeed not exactly symmetric because of the directional search. This directional search (with a stop as soon as the required conditions are not met) is used to avoid finding "consistent" periods that include subperiods with systematically nonsatisfactory behavior.

If in your example, data point 22 corresponds with a nonsatisfactory model result, a search that starts from data point 20 towards the right will stop there (as only 2 successful points on 3 is not enough to meet the required 5 % degree of tolerance). When approaching the data towards the left (starting in data point 60), it is possible that there were enough satisfactory results between data points 60 and 23 in order to overcome the nonsatisfactory behavior of data point 22 with respect to the 5 % degree of tolerance, facilitating a reach towards point 20.

Changes in text: In Section 2.2.5 (Step 5: Assessment of the bidirectional reach for all degrees of tolerance), the directional search is explicitly mentioned by adding a sentence (p.7, lines 30 – 31):

> The temporal span for which this model behaves satisfactory is assessed both in the direction of the previous and the following data points using a directional search, that stops as soon as the required conditions are not met.

**6:** *Why do you consider the 0% tolerance in the BReach plots? Would it not be more correct to set the minimum tolerance at 5% since you used 95% C.I. to define an interval for the measurement errors?*

Author's response: Although a 0 % degree of tolerance is indeed not a realistic condition, the differences between a 0 % and a 5 % plot are informative with regard to the spread of the nonsatisfactory model results. These differences can differ strongly for different data series (e.g. Figure 4a vs. Figure 7a).

Changes in text: none.

**7:** *The figures are not well explained by the information given in the legend and the figure captions. The legend with the grey shades could have "Tolerance" or "Tolerance level" as a title. In the caption it would also be nice to add one or two sentences that explain the plot briefly. Something like "The shaded areas below and above the bisector shows the left and the right reach (vertical axis) of each data point horizontal axis). The tolerance levels indicate the maximum tolerated ratio of data points for which one (or more) rating curve models are outside the measurement uncertainty".*

Author's response: Adapted in Fig. 2-9, 11 and 12.

Changes in text: The legend title is included in Fig. 2-9, 11 and 12. The following sentences are added in the caption of these figures:

For each index in the x axis the gray area indicates the span between the index of the maximum left reach (under the bisector) and the maximum right reach (above the bisector). Each gray tint represents a different degree of tolerance (i.e. percentage of data points allowed to have nonacceptable model results).

**8:** *The grey-shades used in the plots do not correspond to the grey-shades in the legend. In particular, the 0% error bar seems to be black in the plot and a grey-shade in the legend.*
Author's response: Adapted in Figures 2-9, 11 and 12.
Changes in text: Adapted in Figures 2-9, 11 and 12.

**9:** *Standard regression analysis also provide tools for analyzing residuals and in particular other methods are available for detecting step-changes in ordered data. It would be interesting to compare your results to existing methods.*
Author's response: cfr. Response 2 to reviewer #1
Changes in text: cfr. Response 2 to reviewer #1